

# SnowClim v1.0: High-resolution snow model and data for the western United States

Abby C. Lute[1,2], John Abatzoglou[3], Timothy Link[4]

[1]Water Resources Program, University of Idaho, Moscow, ID, 83844, USA
[2]Now at Geophysical Fluid Dynamics Laboratory, National Oceanic and Atmospheric Administration / Program in Atmospheric and Oceanic Sciences, Princeton University, Princeton, NJ, 08540, USA
[3]Management of Complex Systems, University of California, Merced, CA, 95343, USA
[4]Department of Forest, Rangeland, and Fire Sciences, University of Idaho, Moscow, ID, 83844, USA

*Correspondence to*: A.C. Lute (aby.lute@gmail.com)

**Abstract.** Seasonal snowpack dynamics shape the biophysical and societal characteristics of many global regions. However, snowpack accumulation and duration have generally declined in recent decades largely due to anthropogenic climate change. Mechanistic understanding of snowpack spatiotemporal heterogeneity and climate change impacts will benefit from snow data

products that are based on physical principles, that are simulated at high spatial resolution, and that cover large geographic domains. Existing datasets do not meet these requirements, hindering our ability to understand both contemporary and changing snow regimes and to develop adaptation strategies in regions where snowpack patterns and processes are important components of Earth systems.

We developed a computationally efficient physics-based snow model, SnowClim, that can be run in the cloud. The model was

evaluated and calibrated at Snowpack Telemetry sites across the western United States (US), achieving a site-median root mean square error for daily snow water equivalent of 62 mm, bias in peak snow water equivalent of -9.6 mm, and bias in snow duration of 1.2 days when run hourly. Positive biases were found at sites with mean winter temperature above freezing where the estimation of precipitation phase is prone to errors. The model was applied to the western US using newly developed forcing data created by statistically downscaling pre-industrial, historical, and pseudo-global warming climate data from the

Weather Research and Forecasting (WRF) model. The resulting product is the SnowClim dataset, a suite of summary climate and snow metrics for the western US at 210 m spatial resolution (Lute et al., 2021). The physical basis, large extent, and high spatial resolution of this dataset will enable novel analyses of changing hydroclimate and its implications for natural and human systems.

## 1 Introduction

Seasonal snowpack shapes the climatic, hydrologic, ecological, economic, and cultural characteristics of many global regions. Snow is an important determinant of the surface energy balance through its effect on land surface albedo, partitioning of sensible and latent heat fluxes, near-surface atmospheric stability, and horizontal energy transport (Cohen, 1994; Rudisill et al., 2021; Stiegler et al., 2016). Hydrologic benefits of snow include natural water storage, delayed runoff, and cooler stream





temperatures (Bales et al., 2006; Luce et al., 2014). Ecologically, seasonal snow insulates flora and snow-dependent fauna,
controls mobility and foraging opportunities, mediates nutrient cycling, and supplements plant-water availability (Formozov,
1964; Grippa et al., 2005; Jones, 1999). Economically, seasonal snow synchronizes water supply and demand enabling crop
irrigation, fuels a multibillion dollar winter recreation industry in the United States (US) alone, and can cause transportation
delays and accidents (Burakowski and Magnusson, 2012; Qin et al., 2020; Seeherman and Liu, 2015; Sturm et al., 2017).
Finally, seasonal snow is a defining aspect of many cultures globally, shaping language, traditions, and sense of self (Eira et
al., 2013; Mergen, 1997).

In many mountain regions, recent decades have seen less precipitation falling as snow, lower peak snow water equivalent
(SWE), shorter snow duration, and earlier snowmelt runoff (Choi et al., 2010; Fritze et al., 2011; Knowles et al., 2006; Mote
et al., 2018). These developments are expected to continue in the coming decades, resulting in substantial declines (>50%) in
seasonal snowpack for areas such as the western US and significant impacts to human and natural systems (Fyfe et al., 2017;
Huss et al., 2017; Marshall et al., 2019a; Siirila-Woodburn et al., 2021). In addition to these macroscale developments, there
are important nuances to changing snow. Increased atmospheric water vapor due to warming is expected to enable larger
snowfall events (Lute et al., 2015), which may buffer declines in snowpack (Marshall et al., 2020). Changes in atmospheric
circulation may affect snow accumulation, for example by diminishing orographic precipitation enhancement (Luce et al.,
2013) or altering characteristics of atmospheric rivers (Dettinger, 2011). Decreasing snow cover will result in increased
hydrologic importance of microclimates that serve as snow refugia, such as high elevations, deposition zones, and shaded areas
(Marshall et al., 2019b; McLaughlin et al., 2017). A warmer and moister atmosphere will shift the relative importance of
snowpack energy and mass budget terms, resulting, for example, in slower snowmelt (Musselman et al., 2017), changes to the
partitioning of snow ablation between runoff and sublimation (Sexstone et al., 2018), and increasing rain-on-snow risk in
regions that retain snow cover (Musselman et al., 2018).

Understanding these changes and their implications often requires snow models and modeled snow data products (hereafter
snow data) that satisfy at least one of several criteria. These criteria include that the data is: a) simulated with physics-based
representations of energy and mass transfer processes, b) spatially continuous, c) high spatial resolution, d) large extent, e)
multivariate, and f) multitemporal. To address some questions about contemporary or future snow, the snow models themselves
are needed and must be able to synthesize data that satisfies these criteria. Snow data developed from physical principles is
argued to be necessary for both capturing the spatial variability of energy fluxes across the landscape and providing physically
realistic simulations of the effects of climate change (Kumar et al., 2013; Raleigh and Clark, 2014). To assess changes in
snowpack across a landscape, spatially continuous data are needed. In areas of complex terrain, high spatial resolution data are
necessary to resolve the effects of elevation and shading (Barsugli et al., 2020; Sohrabi et al., 2019; Winstral et al., 2014). For
some applications, such as water management and species distribution modeling, snow data may need to cover large geographic
domains. Multiple snow metrics are needed for diverse applications (e.g., SWE for water management, snow depth for
wildlife). Finally, historical and future data are necessary to evaluate changes over time and to inform long term planning and
development of adaptation strategies for specific locales.





There are two major hurdles to the development of a snow dataset that meets all of these criteria: appropriate forcing data and computational cost. Presently, large-extent climate datasets only achieve horizontal resolutions of up to 1 km (e.g. Abatzoglou

and Brown, 2012; Fick and Hijmans, 2017; Thornton et al., 2020) and the finer resolution datasets cover limited domains or are restricted to historical periods (Dietrich et al., 2019; Holden et al., 2011, 2016). Second, even with appropriate forcing data, the computational expense of running snow models has generally forced the selection of some of these criteria at the expense of others (Winstral et al., 2014). For example, a temperature-index model might be used for applications requiring rapid results over large domains (e.g. SNOW-17; Anderson, 2006), a physics-based model might be run at high resolution over watershed

sized domains (Garen and Marks, 2005; Liston and Elder, 2006), or a physics-based model might be run at coarser resolution over a larger extent (e.g. SNODAS, National Operational Hydrologic Remote Sensing Center, 2004; WRF, Rasmussen and Liu, 2017; Gergel et al., 2017; Wrzesien et al., 2018). There is potential for clever computational solutions and model formulations, such as variable resolution grids, to alleviate these trade-offs to some extent (Marsh et al., 2020).

In this study we developed a computationally efficient physics-based snow model called SnowClim that has a flexible model

structure and can be run in the cloud (Lute et al., 2021). The model retains the most important components of physically based models, including the complete energy balance and internal snowpack energetics, while omitting more computationally expensive components such as horizontal transport, multiple layers, and iterative solutions for snow surface temperature. Unlike existing models, this simplified physics-based model is efficient enough to be run over sub-continental domains at high spatial resolution. We force the SnowClim model with pre-industrial (1850-1879), historical (2000-2013), and projected future

(2071-2100) meteorological data from the Weather Research and Forecasting (WRF) model downscaled to correct for terrain effects. We then applied the model to the western US to create the SnowClim dataset, a multivariate, gridded, snow and climate dataset for three time periods at 210m spatial resolution. Here we provide a description of the model and its application to the western US, including parameterization, calibration, climate forcing data preparation, and resultant datasets.

## 2 Model Description

### 2.1 Model Overview

The SnowClim model is a fully distributed energy and mass balance snow model. It simulates the snowpack as a single layer, but accounts for different surface and pack temperatures (Fig. 1). The effects of vegetation, fractional snow cover, and snow redistribution via gravitational and wind-driven processes are not represented.

The model has a flexible structure to facilitate uncertainty analysis and application to new conditions. This flexible structure

includes tunable parameters, customizability of the spatiotemporal application, and process modularity. Key parameters (Table 2) are user-defined as opposed to hard-coded in the model, allowing for calibration of the model to new conditions and regions as seen fit. The temporal and spatial resolution and extent are also user-defined, which allows users to adjust to computational





constraints and the requirements of the project. Finally, key processes such as albedo and turbulent fluxes are modularized to allow evaluation of alternative process representations.

The required forcings are described in Table 1. The model is written in MATLAB and the code is available at https://www.hydroshare.org/resource/dc3a40e067bf416d82d87c664d2edcc7/. The model can be run in the cloud using MATLAB Online through the HydroShare Platform hosted by the Consortium of Universities for the Advancement of Hydrologic Science, Inc. (CUAHSI).

## 2.2 Energy Balance

The SnowClim model evaluates the surface energy balance at each time step such that

$$Q_{net} = SW_\downarrow - SW_\uparrow + LW_\downarrow - LW_\uparrow + H + E_i + E_w + P + G \qquad (1)$$

where $Q_{net}$ is the net surface energy flux, $SW_\downarrow$ is the downward shortwave radiation at the surface, $SW_\uparrow$ is the upward shortwave radiation at the surface, $LW_\downarrow$ is the downward longwave radiation at the surface, $LW_\uparrow$ is the upward longwave radiation at the surface, $H$ is the sensible heat flux, $E_i$ and $E_w$ are the latent heat fluxes of ice and water, $P$ is the advected heat

flux from precipitation, and $G$ is the ground heat flux (Fig. 1).

### 2.2.1 Shortwave Radiation

Upward shortwave radiation is equivalent to

$$SW_\uparrow = SW_\downarrow \alpha \qquad (2)$$

where $\alpha$ is the spectrally integrated snow surface albedo.

Springtime snow model simulations are sensitive to the specific albedo algorithm (Etchevers et al., 2004; Günther et al., 2019). The SnowClim model provides three options for computing snow albedo (*albedo_opt*). In all options, albedo decays with time and the albedo of shallow snowpacks (<100 mm depth) is diminished to account for the albedo of the ground surface, assumed to be 0.25 (Walter et al., 2005). A user-specified maximum albedo parameter (*albedo_max*) is used in each method.

The simplest albedo model (Essery et al., 2013; hereafter Essery), is empirical and sets albedo decay as a function of snowpack

temperature. Snow albedo is augmented based on the occurrence and amount of new snow. Parameters other than the maximum albedo (minimum albedo, new snow threshold, linear and exponential albedo decline rates) are taken from Douville et al., (1995).

In the second albedo model (Hamman et al., 2018; Liang et al., 1994; hereafter VIC), snowpacks with new snow depth > 10 mm and non-zero cold content receive the maximum snow albedo. Other albedo parameters are taken directly from VIC. Snow

albedo decays more rapidly for melting snowpacks than cold snowpacks (cold content, $cc < 0$).

The final albedo model (Tarboton and Luce, 1996; hereafter Tarboton) accounts for the wavelength dependence of albedo by computing separate visible and near-infrared band albedos as a function of snow surface age and solar illumination angle. The



maximum albedo parameter is set equal to the average of the maximum visible band and infrared band albedos. This is the only albedo model of the three that includes a correction for illumination angle.

### 2.2.2 Longwave Radiation

Upward longwave radiation is a function of snow surface temperature ($T_s$) in degrees Celsius, snow emissivity ($\varepsilon$), and the Stefan-Boltzmann constant ($\sigma$) such that

$$LW_\uparrow = \varepsilon\sigma(T_s + 273.15)^4 \tag{3}$$

We assume $\varepsilon = 0.98$ (Armstrong and Brun, 2008). We consider $T_s$ to be a function of the dewpoint temperature ($T_d$; Raleigh et al., 2013), such that

$$T_s = min(0\ °C, T_d + T_{add}) \tag{4}$$

where $T_{add}$ is an augmentation parameter that increases $T_s$ and improves simulations of sublimation. For further discussion of $T_s$ see section 2.2.6.

### 2.2.3 Turbulent Fluxes

The turbulent fluxes, $H$, $E_i$, and $E_w$, are estimated using a Richardson number parameterization of the exchange coefficient following Essery et al., (2013). The bulk formula are

$$H = -\rho_a c_a C_H U_a (T_s - T_a) \tag{5}$$

$$E_i = -\rho_a C_H U_a (Q_s - Q_a)\lambda_s \quad for\ T_s < 0 \tag{6}$$

$$E_w = -\rho_a C_H U_a (Q_s - Q_a)\lambda_v \quad for\ T_s = 0 \tag{7}$$

where $\rho_a$ is the air density, $c_a$ is the specific heat capacity of air, $C_H$ is the bulk exchange coefficient that accounts for near-surface atmospheric stability, $U_a$ is the wind speed, $Q_s$ is the specific humidity of the snow surface, and $Q_a$ is the specific humidity of the air which is a required forcing. The specific humidity of the snow surface is calculated from $T_s$. The exchange coefficient $C_H$ is parameterized as a function of the near-surface atmospheric stability as captured by the bulk Richardson number ($Ri_B$) such that

$$C_H = F_H(Ri_B)C_{HN} \tag{8}$$

$$Ri_B = (gz_u(T_a - T_s))/(T_a U_a^2) \tag{9}$$

$$C_{HN} = k^2[ln(z_u/z_0)]^{-1}[ln(z_T/z_h)]^{-1} \tag{10}$$

$$F_H(Ri_B) = 1 \qquad for\ Ri_B = 0 \tag{11}$$

$$F_H(Ri_B) = 1 - (3cRi_B)/(1 + 3c^2 C_{HN}(-Ri_B z_u/z_0)^{1/2}) \quad for\ Ri_B < 0 \tag{12}$$

$$F_H(Ri_B) = [1 + (2cRi_B)/(1 + Ri_B)^{1/2}]^{-1} \qquad for\ Ri_B > 0 \tag{13}$$





where $g$ is gravitational acceleration, $z_u$ is the height of simulated wind speeds, $z_T$ is the height of simulated air temperatures, $z_0$ is the surface roughness length for momentum, $z_h$ is the surface roughness length for heat and water vapor, and c is a constant assumed to equal 5 (Louis, 1979). $z_0$ and $z_h$ are adjustable user-specified parameters (Table 2).

An optional windless exchange coefficient is available to counter large radiative losses particularly during stable conditions (Helgason and Pomeroy, 2012; Jordan, 1991). Application of the windless exchange coefficient can be modified through three parameters: *E0_value*, *E0_app*, and *E0_stability* (Table 2). *E0_value* is the value in W m⁻² of the windless exchange coefficient. *E0_app* controls the application of the windless heat exchange coefficient to the sensible and latent heat fluxes; an *E0_app* value of 1 applies the coefficient only to the sensible heat flux, whereas an *E0_app* value of 2 applies the coefficient to both the sensible and latent heat fluxes. *E0_stability* controls the type of conditions where the windless coefficient is applied; an *E0_stability* value of 1 applies the coefficient to all conditions, whereas an *E0_stability* value of 2 applies the condition only under stable atmospheric conditions.

### 2.2.4 Precipitation heat flux

The heat flux of liquid precipitation is

$$P = c_w \rho_w T_d P_{rain} \tag{14}$$

where $c_w$ is the specific heat of water, $p_w$ is the density of water, and $P_{rain}$ is the rate of liquid precipitation. The heat flux of solid precipitation ($S$) is handled separately for diagnostic purposes and is added directly to the snowpack cold content.

$$S = c_i \rho_w T_d P_{snow} \tag{15}$$

where $c_i$ is the heat capacity of ice and $P_{snow}$ is the rate of snowfall.

### 2.2.5 Ground heat flux

The ground heat flux can be important in controlling the onset of seasonal snow accumulation, particularly in warmer environments (e.g., Mazurkiewicz et al., 2008). However, under most circumstances $G$ is thought to provide a minor contribution to the energy budget (DeWalle and Rango, 2008). In the interest of model efficiency and to avoid uncertainties associated with estimating soil temperatures and thermal conductivities, we use a constant $G$ of 2 W m⁻² (Walter et al., 2005), similar to other models (Etchevers et al., 2002).

### 2.2.6 Enhanced single layer approach

Single layer snow models typically provide less physically realistic snowpack simulations than multilayer models due to their simplified treatment of energy transfer within the snowpack (Blöschl and Kirnbauer, 1991; Waliser et al., 2011). Bulk single layer conceptualizations treat the surface temperature and energy balance as synonymous with the pack temperature and energy balance, ignoring the contrast between the thin surface layer which is highly sensitive to the near-surface atmosphere, and the





pack, which is characterized by thermal inertia, i.e., cold content. These distinctions are key to accurate modeling of snowpack

heat fluxes (Blöschl and Kirnbauer, 1991) and snowpack ablation (Waliser et al., 2011).

 To address these shortcomings, advanced single layer snow models have differentiated between surface and pack

temperatures, while attempting to maintain the parsimony of a single layer model (Tarboton and Luce, 1996; You et al., 2014).

However, these approaches typically require iterative methods to solve for snow surface temperature that can be

computationally expensive (Wigmosta et al., 1994) and subject to large uncertainty (Raleigh et al., 2013).

 The present model uses a two-step modification of the net surface energy flux to approximate the conduction of energy

between the surface and the snowpack. This approach enables separate temperatures and energy balances for surface and pack

components while retaining the computational efficiency necessary to accomplish the modeling objectives of both large spatial

extent and relatively fine resolution. In this approach, the surface is conceptualized as a skin with zero depth.

First, we apply a temporal running mean to the net surface energy flux to approximate the attenuation with depth of the

characteristic diurnal variations in energy at the surface, akin to the approach taken by You et al., (2014). The smoothed energy

flux from the surface to the pack at each time step ($\overline{Q_{net}}$) is calculated as the average net surface energy flux over a period

*smooth_hrs*, that is a tunable parameter (Table 2). This approach reduces unrealistic high frequency modifications of the cold

content and large amplitude freeze-thaw cycles during the ablation season.

Second, we apply a progressive tax on the negative net energy flux to the snowpack to limit the excessive accumulation of

cold content that results from all surface energy being directly translated to the pack. The net effect of the energy tax is to

reduce snowpack cold content, resulting in more accurate cold content simulations similar to those from other, more complex

physics-based models (Jennings et al., 2018a; not shown). Other single layer models have sought to limit cold content, however

they used approaches that required site specific calibration (Blöschl and Kirnbauer, 1991; Braun, 1984). We apply a

progressive tax such that negative energy fluxes to snowpacks with larger cold content receive larger taxes:

$$Q_{pack} = \overline{Q_{net}} \qquad\qquad for\ \overline{Q_{net}} \geq 0$$
$$Q_{pack} = \overline{Q_{net}} \times (1 - tax) \qquad for\ \overline{Q_{net}} < 0 \qquad\qquad\qquad (16)$$
$$tax = \frac{cc - cc_0}{cc_1} \times maxtax\ \ such\ that\ 0 \leq tax \leq maxtax$$

$\overline{Q_{net}}$ is the smoothed net surface energy flux, $Q_{pack}$ is the energy flux from the surface to the pack, and $cc$ is the snowpack

cold content. $cc_0$, $cc_1$, and $maxtax$ are tunable parameters that define the maximum (least negative) cold content to which the

tax should be applied, the range of cold content over which the tax should be applied ($cc_0$ to $cc_0 + cc_1$), and the maximum

possible tax, respectively (Table 2). Negative energy fluxes to snowpacks with cold contents less negative than $cc_0$ receive 0

tax, and negative energy fluxes to snowpacks with cold contents more negative than $cc_0 + cc_1$ receive a tax equal to $maxtax$.

$Q_{pack}$ is added to the snowpack cold content ($cc$) at each time step. Pack temperature ($T_{pack}$) can be obtained from cold

content:

$$T_{pack} = cc\ /\ (\rho_w \times c_i \times SWE) \qquad\qquad\qquad\qquad\qquad (17)$$

where $SWE$ is the snow water equivalent.





### 2.2.7 Modification for shallow snowpacks

We developed a computationally efficient approach for controlling energy balance instabilities for shallow snowpacks. Marks
et al., (1999) addressed the problem by shifting to progressively smaller time steps. In the interest of computational efficiency,
we take an alternative approach. When modeled SWE is less than a threshold value, $T_{pack}$ is set equal to the minimum of $T_a$
and 0°C. Cold content is then updated according to this new temperature. The threshold for applying this correction is 15 mm
of SWE for every hour in the time step (e.g. for a model run at a 4 hour time step the temperature correction would be applied
to snowpacks with 60 mm SWE or less). Constraining $T_{pack}$ and cold content in this way is reasonable given that surface and
pack temperatures are likely to be similar for shallow snowpacks and the strong correspondence between $T_s$ and $T_a$ (Helgason
and Pomeroy, 2012).

### 2.3 Mass Balance

The mass balance of the solid and liquid portions of the snowpack are evaluated at each time step as

$$M_s = M_{snow} + M_{ref} - M_{melt} + M_{dep} - M_{sub} \tag{18}$$

$$M_l = M_{rain} - M_{ref} + M_{melt} - M_{runoff} + M_{cond} - M_{evap} \tag{19}$$

where $M_s$ is the mass of the solid portion of the snowpack, $M_{snow}$ is the mass of new snowfall, $M_{ref}$ is the mass of liquid water
in the snowpack that has been refrozen, $M_{melt}$ is the mass of snow that has melted, $M_{dep}$ is the mass of deposition, $M_{sub}$ is the
mass of sublimation, $M_l$ is the mass of the liquid in the snowpack, $M_{rain}$ is the mass of rain added to the snowpack, $M_{runoff}$
is the mass of liquid water that has left the snowpack as runoff, $M_{cond}$ is the mass of condensation, and $M_{evap}$ is the mass of
evaporation (Fig. 1).

### 2.3.1 Accumulation

Snowfall is calculated as an air temperature and relative humidity dependent fraction of precipitation using the bivariate logistic
regression model of Jennings et al., (2018b). We use a non-binary formulation to allow for mixed phase precipitation. New
snowfall amounts less than 0.1 mm water equivalent per hour are set to 0. Rainfall is the difference between precipitation and
snowfall. The temperature of new snowfall is set equal to the minimum of the dewpoint temperature and freezing point (0°C)
whereas the temperature of rainfall is set equal to the maximum of the dewpoint temperature and the freezing point (Marks et
al., 2013; Raleigh et al., 2013).
The density of new snowfall is calculated as a function of air temperature (Anderson, 1976) using constants identified by
Oleson et al., (2004). Compaction of the snowpack is modeled as a function of SWE and snowpack temperature following
Anderson, (1976) and using constants from Boone, (2002) for the ISBA-ES snow model. Snow depth is a function of SWE
and density and is updated following changes in either variable.





### 2.3.2 Melt

Positive net energy flux must satisfy the snowpack cold content before melt can occur. Melt is equivalent to the minimum of the current SWE and the potential melt,

$$melt_{pot} = Q_{pack}/(\lambda_f \times \rho_w) \quad \text{for} \quad Q_{pack} > 0 \tag{20}$$

where $\lambda_f$ is the latent heat of freezing.

### 2.3.3 Liquid water content

Rainfall, melt, and condensation are added to and evaporation is subtracted from the snowpack liquid water content. Snowpack liquid water content in excess of the liquid water holding capacity of the snowpack contributes to runoff. The liquid water

holding capacity of the snowpack is the product of snow depth and the maximum liquid water fraction (*lw_max*, Table 2). Liquid water content below this threshold but greater than the minimum liquid water content (equivalent to 1% of snow depth (Marsh, 1991)) is allowed to drain at a rate of 100 mm h[-1] (based on values in Armstrong and Brun, 2008; DeWalle and Rango, 2008).

### 2.3.4 Refreezing

Excess cold content can be used to refreeze liquid water in the snowpack. The amount of water refrozen is the minimum of the total liquid water content of the snowpack and the potential refreezing,

$$refreeze_{pot} = -cc \ /(\lambda_f \times \rho_w) \quad \text{for } cc < 0 \tag{21}$$

$$M_{ref} = min(refreeze_{pot}, M_l) \qquad \text{for } cc < 0 \tag{22}$$

Energy released by refreezing is added to the snowpack cold content and the refrozen mass is added to the SWE, increasing

the snowpack density (we assume no change in snow depth).

### 2.3.5 Sublimation and condensation

Latent heat transfer results in sublimation or evaporation from or deposition or condensation onto the snowpack, such that

$$M_{sub} = -E_i \ /(\lambda_s \times \rho_w) \quad \text{for} \quad E_i < 0 \ and \ T_s < 0 \tag{23}$$

$$M_{evap} = -E_w \ /(\lambda_v \times \rho_w) \quad \text{for} \quad E_w < 0 \ and \ T_s = 0 \tag{24}$$

$$M_{dep} = -E_i \ /(\lambda_s \times \rho_w) \quad \text{for} \quad E_i > 0 \ and \ T_s < 0 \tag{25}$$

$$M_{cond} = -E_w \ /(\lambda_v \times \rho_w) \quad \text{for} \quad E_w > 0 \ and \ T_s = 0 \tag{26}$$

where $\lambda_s$ is the latent heat of sublimation and $\lambda_v$ is the latent heat of vaporization.





**3 Model Application to the Western United States**

The SnowClim model was evaluated and calibrated at a collection of automated snow stations across montane portions of the
western US and further applied to the broader western US to create the SnowClim dataset. We describe the preparation and
downscaling of the meteorological forcing data, the model calibration, and the model simulations for the western US. The
model was calibrated at Snowpack Telemetry (SNOTEL) sites and model performance at these sites was used to select the
parameters and temporal resolution at which to run the model over the full domain.

**3.1 Spatial resolution**

To balance the competing ambitions of high spatial resolution and computational feasibility over the western US domain, we
used variable spatial resolutions. Regions of complex terrain were modeled at 210 m (hereafter 'fine'). This high resolution
enhances the model's ability to capture the effects of elevation, aspect, and slope on snowpack in complex terrain. Regions of
less complex terrain were modeled at 1050 m (hereafter 'coarse'). Terrain complexity was assessed for each coarse grid cell
by examining the elevations and downscaled solar radiation values for the 25 collocated fine grid cells. If the elevation
difference across the fine cells was less than 50 m and the maximum percent difference in solar radiation was less than 10%,
then snow simulations were completed at coarse resolution. Otherwise, simulations were completed at fine resolution. This
resulted in approximately 30% of the domain being modeled at coarse resolution (Fig. A1). Grid cells were defined using the
1 arc-second National Elevation Dataset Digital Elevation Model (DEM; Gesch et al., 2018), aggregated to 210 m or 1050 m.

**3.2 Forcing data preparation**

Hourly meteorological data from the Weather Research and Forecasting model (WRF; Rasmussen and Liu, 2017) were
downscaled to force the snow model (Table 3). Forcing data was developed for a historical period, future period, and pre-
industrial period. The raw WRF data consisted of 4 km spatial resolution hourly simulations for 1 October 2000 to 30
September 2013, that used initial and boundary conditions from ERA-Interim (Dee et al., 2011), herein referred to as the
historical period. A pseudo-global warming run was also performed by perturbing ERA-Interim by average differences from
a suite of climate models participating in the Fifth Coupled Model Intercomparison Project (CMIP5; Taylor et al., 2012)
between 1976-2005 and 2071-2100 under the RCP 8.5 scenario (Rasmussen and Liu, 2017). Pre-industrial forcing data was
developed by perturbing the downscaled historical WRF data by monthly climatological differences in climate between pre-
industrial (1850-1879) and the historical period using a pattern scaling approach (Mitchell, 2003) based on spatially varying
differences in variables from the CMIP5 models.

Spatial downscaling for all variables except solar radiation was accomplished using moving window lapse rates (i.e., the
change in the variable with elevation). Lapse rate downscaling has been shown to perform well relative to other statistical
downscaling approaches in mountainous terrain (Praskievicz, 2018; Wang et al., 2012). We estimated monthly lapse rates for





each grid cell and each variable, except for temperature for which we estimated hourly lapse rates for each grid cell. Windows of 7x7 WRF grid cells, or 28 km x 28 km, were used to balance the competing objectives of sufficient data points and the

ability to capture local phenomena (Lute and Abatzoglou, 2021). Lapse rate corrections were applied hourly using the elevation difference between the WRF grid cell and the target DEM grid cell. For air pressure, lapse rates were calculated from and applied to temporally averaged WRF data. Grid cells not classified as land by WRF were excluded from lapse rate calculations. For precipitation, a modified version of the methods above was used. Prior to calculating lapse rates, WRF precipitation was bias-corrected to monthly 4 km precipitation from PRISM (PRISM Climate Group, 2015) by calculating monthly correction

ratios- the ratio of total monthly PRISM precipitation to total monthly WRF precipitation. Correction ratios were set to 1 (no correction) when monthly WRF precipitation was 0 or when the ratio was infinite. Monthly precipitation lapse rates were divided by the number of hours with precipitation each month in the underlying WRF data and days with 0 precipitation were maintained in the downscaled data to avoid precipitation everyday due to non-zero monthly lapse rates.

Solar radiation was downscaled to the target DEM using the insol package in R (Corripio, 2015) following the approach of

Lute and Abatzoglou, (2021) which preserves the atmospheric effects (e.g. cloud cover) captured by WRF and also accounts for slope, aspect, self-shading, and shading by adjacent terrain. Parameters required by the algorithm, including visibility, RH, and temperature, were assumed to be constant. Terrain corrections were calculated for the midpoint of each hour of the middle day of each month, aggregated to the desired temporal resolution using a weighting scheme based on the amount of solar radiation each hour, and then interpolated to the full time period.

For model calibration at SNOTEL sites (see next section), the above downscaling procedures were applied, but values were adjusted based on the elevation difference between the SNOTEL site and the collocated WRF grid cell based on calculated lapse rates. Downscaled WRF precipitation was bias-corrected to SNOTEL sites by applying a monthly correction factor consisting of the ratio of the total SNOTEL precipitation to the total WRF precipitation similar to Havens et al., (2019). We note that such bias correction approaches may not address issues of precipitation undercatch at SNOTEL sites.

Additional variables needed to force the snow model including specific humidity, relative humidity, and dewpoint temperature were derived from the downscaled water vapor mixing ratio, air temperature, and air pressure data using standard methods. Dewpoint temperatures exceeding the air temperature were set equal to the air temperature.

### 3.3 Model calibration

### 3.3.1 Calibration methods

The model was calibrated at SNOTEL sites across the mountains of the western US to select a single best parameter set across all sites. A total of 170 SNOTEL sites were selected meeting the following requirements:

1. elevation difference of less than 75 m relative to the collocated WRF grid cell;
2. missing no more than 1% of daily precipitation and SWE observations between October and May in every water year between 1 October 2000 and 30 September 2013;





3.   located more than 25 km from any other SNOTEL site.

Missing SWE values were infilled using linear interpolation. Missing precipitation values were infilled using an inverse distance weighted average of the values at the three closest sites.

Calibration consisted of running the model across all SNOTEL sites for each possible combination of parameters listed in Table 2. Model performance was assessed using the mean absolute percent error (MAPE) of annual maximum SWE (maxswe),

the MAPE of annual snow duration, and the root mean squared error (RMSE) of daily SWE at each site. Snow duration was defined as the duration (in days) of the longest period of consecutive days with SWE > 0. RMSE was computed for days when observed SWE exceeded 10 mm. Additionally, we used the mean error (ME) and mean percent error (MPE) of maxswe and duration to visualize calibration errors. The optimal parameter set was selected using Pareto preference ordering (Khu and Madsen, 2005) based on the median of each statistic across stations.

The model was subsequently evaluated for different run time steps (1, 2, 3, 4, 6, 8, 12, and 24 hours). Separate model calibration for each time step selected similar parameter sets to the hourly run, so the hourly parameter set was used for all time steps. Model performance was again assessed as described above.

### 3.3.2 Calibration results

Snow model calibration via Pareto optimization selected a single best parameter set (Table 2). The station median MAPE of

maxswe, MAPE of snow duration, and daily RMSE for this parameter set were 15.6%, 8.86%, and 62.2 mm, respectively. The spatial distribution of ME and MPE in maxswe and duration lacked strong coherent spatial patterns (Fig. 2) and spatial correlations ($R^2$) for a variety of snow metrics exceeded 0.75 (Table 4), suggesting that the model captured major climate related effects and sources of large-scale spatial snow variability. The largest negative biases were found at drier sites with relatively shallow or intermittent snowpacks (Fig. A2). The largest positive biases were found at sites with mean winter

temperatures at or above freezing, where snow accumulation is very sensitive to the partitioning of precipitation into rain vs. snow (Fig. A2). A time series of observed and modeled SWE at one site with error values close to the station median values illustrates the model performance on a daily scale (Fig. 3). The model also captured key components of interannual snowpack variability over the short historical period; the station median correlation coefficients for maxswe and for snow duration were 0.92 and 0.69, respectively. The station correlations did not demonstrate any clear geographic or climatic patterns. This lends

confidence to the model's ability to accurately simulate snow dynamics in other climates. The parameter sensitivity of the model is shown in Fig. A3.

Model performance deteriorated as temporal resolution coarsened from 1 hour to 12 hours, but improved slightly for the 24 hour timestep (Fig. 4). Model performance was somewhat sensitive to the hours used for aggregation; other aggregation windows showed continuous performance deterioration with coarsening temporal resolution (not shown). A timestep of 4

hours was selected for the full western US model run to balance the objectives of computational efficiency and model performance. The station median MAPE of maxswe, MAPE of snow duration, and RMSE for the 4 hour time step were 17.8%,





11.9%, and 75.4 mm, respectively. We note that simulations without the modification for shallow snowpacks (Sect. 2.2.7) degraded more consistently and significantly with coarsening temporal resolution (Fig. A4).

### 3.4 Model results for the Western United States

The SnowClim model was applied to the western US (contiguous US west of 104°W) using the parameters identified above, a temporal resolution of 4 hours, and a variable spatial resolution as described previously (210 m-1050 m horizontal resolution). The model was run in parallel on a high-performance computer with 34 cores and 128GB RAM. The compute time for downscaling the climate forcings and executing the snow model was 10 days and 3.5 days, respectively, for the historical period.

Historical maxswe was 111 mm, spatially averaged across the full western US domain, and locations with historical maxswe < 50 mm were found in the warmer southern and southwestern regions and in the northeastern portion of the domain where winters are relatively dry (Fig. 5a). Under the future scenario, the areas with maxswe < 50 mm expanded to encompass many lower elevation areas and spatially averaged maxswe declined to 53 mm (Fig. 5b). Historical snow duration averaged 101 days, spatially averaged across the full western US domain (Fig. 5c), but declined to 54 days in the future scenario (Fig. 5d).

There were only a handful of locations with increases in maxswe or duration in the future period compared with the historical period, and these increases were small (Fig. 6). The largest relative declines in maxswe and duration were found at low elevations. On average, maxswe decreased by 49% across locations with at least 50 mm maxswe historically and snow duration decreased by 57% across locations with historical snow duration greater than zero. Summaries of historical and future maxswe and duration by 4-digit hydrologic unit code (HUC) are provided in Table A1.

Compared to existing large extent, multitemporal, physics-based snow datasets such as that from the 4 km WRF runs (Rasmussen and Liu, 2017), SnowClim provided a much more nuanced picture of changing snow, particularly in areas of complex terrain. For example, Fig. 7 shows relative changes in maxswe for the Uinta Mountains in northeastern Utah as simulated directly by the 4 km WRF product and by SnowClim. SnowClim captured effects of elevation and aspect, including greater percent reductions in maxswe at lower elevations and on south facing aspects, similar to Barsugli et al., (2020). Nuanced

results such as these are only possible with high-resolution, physics-based snow modeling.

Comparison of SnowClim snow depth with finer resolution observations further highlights some of the strengths and limitations of SnowClim. For example, in the Boulder Creek Watershed, Colorado, SnowClim captured the broad scale spatial patterns of snow depth that are present in depth observations derived from Light Detection and Ranging (LiDAR) (described in Harpold et al., 2014) aggregated from the original 1 m resolution to the resolution of SnowClim on 20 May 2010 (Fig. 8).

However, SnowClim snow depth was less spatially variable, particularly in the higher elevation western portion of the domain. Evaluations such as this are particularly challenging as they simultaneously evaluate the spatial and temporal fidelity of the forcing data, the snow model, and in particular the snow density algorithm. The muted spatial variability in SnowClim can be attributed to a combination of factors, chief among which may be the lack of wind-snow interaction in the current SnowClim





model formulation. Snow redistribution by wind and blowing snow sublimation have a significant effect on snowpack

heterogeneity in this region (Knowles et al., 2012; Sexstone et al., 2018; Winstral et al., 2002); one study showed that a model incorporating wind redistribution captured 8-23% more of the spatial variability in snow depth than a model without these processes (Winstral et al., 2002). In order to better simulate snowpack in windy environments such as the alpine area shown here, incorporation of blowing snow transport and sublimation into future SnowClim model formulations should be considered.

## 4 Discussion and Conclusion

Through the development of a new computationally efficient snow model, SnowClim, and novel forcing data, we have overcome the two major hurdles to achieving snow data that meets the criteria outlined in the introduction. SnowClim's unique balance of mostly physical and some empirical components allows it to capture contrasts in radiative loading in complex terrain, timing and rate of ablation, and responses to future climate, while maintaining computational efficiency. The SnowClim dataset is spatially continuous across the western US at sub-kilometer resolution in complex terrain, enabling both

high-resolution and large-extent analyses. The inclusion of multiple snow variables and compatible climate variables across multiple time periods will empower analyses of hydroclimatic responses to changing climate.

The SnowClim model excludes some processes that might be included in more complex, computationally expensive models, such as vegetation related processes, blowing snow transport and sublimation, and gravitational redistribution. In some contexts, these processes may be necessary for accurate modeling of the snowpack (Freudiger et al., 2017; Musselman et al.,

2008; Pomeroy et al., 1993). As it is, the SnowClim data can be considered a potential snow layer in vegetated areas and is expected to be most realistic in minimally vegetated areas with relatively little snow redistribution. Given the complexity of vegetation-snow processes, incorporation of vegetation effects may add significant computational expense and is hindered by the need for vegetation related data and parameters that are expected to change between the time periods considered here. However, incorporation of an optional vegetation routine to be used when data and computational resources are available is a

logical next step. Approaches for incorporating blowing snow transport either require a) high-resolution wind fields input to semi-empirical or 3D turbulent-diffusion models (summarized in Mott et al., 2018), requiring more sophisticated downscaling of wind fields than what was done here and substantially increasing computational cost, or b) require calibration of terrain parameters (e.g. Winstral et al., 2013), which would be possible, but both challenging and computationally intensive for a large-scale model such as SnowClim. Simple algorithms do exist for modeling gravitational redistribution of snow (Bernhardt

and Schulz, 2010), however incorporation of either blowing snow transport or avalanching would necessitate restructuring of the model as a semi- or fully-distributed model with spatial interaction, which alone would likely reduce the computational efficiency of the model. The model also includes simplified representations of the ground heat flux and snow surface temperature, which may be better captured by more physics-based approaches. In particular, a more nuanced treatment of the ground heat flux may be desired in warmer snow climates (Mazurkiewicz et al., 2008).





Contrasts between modeled and observed snow metrics stem from several factors, including but not limited to: uncertainties in climate forcings, SNOTEL site specific factors that the model neglects such as fine scale topographic and vegetation patterns, and errors in model specification including process representation and calibration. Despite these factors, errors at SNOTEL sites from the hourly SnowClim model run were relatively small and compared well with errors reported for other gridded snow products. Ikeda et al., (2021) evaluated the snow simulations from the same 4 km WRF model runs that we

sourced our raw climate forcings from (Rasmussen and Liu, 2017). Relative to SNOTEL sites, they found a -26.2% bias in maxswe. In contrast, the SnowClim model achieves a maxswe bias of only 0.15%. Wrzesien et al., (2018) compared maxswe at SNOTEL sites to maxswe from 9 km WRF simulations. Across sites, they found a correlation coefficient of 0.55 and a bias of -89 mm. SnowClim achieves a correlation coefficient of 0.94 and bias of -11 mm. In the Sierra Nevadas, Guan et al., (2013) blended modeled, remotely sensed, and observed data to capture SWE at 6 sites. Their method achieved a SWE RMSE of 205

mm compared to snow surveys. The SnowClim mean RMSE of daily SWE was 77 mm across all sites and 166 mm at Sierra Nevada sites. While errors at SNOTEL sites were generally low, the model did tend to overestimate maxswe and duration at some warm/wet sites and underestimate these metrics at dry sites (Fig. A2). Further evaluation of the parameters used here in more marginal snow environments would lend additional confidence to the application of SnowClim data in these areas. While the model's excellent performance relative to SNOTEL observations is in part due to the fact that the model was calibrated to

SNOTEL data, the model could easily be calibrated to other observations for application in other contexts.

The flexible, modularized, structure of the SnowClim model lends itself to calibration, parameter sensitivity assessment, and experimentation. In the western US, model performance was particularly sensitive to the choice of albedo algorithm and snow surface temperature parameterization, in line with previous findings (Etchevers et al., 2004; Günther et al., 2019; Slater et al., 2001; Fig. A3). Given the importance of impurities (e.g. tree litter, dust, and black carbon) on snow albedo and consequently

snow melt (Waliser et al., 2011), a future step will be to add albedo algorithms that account for these effects. The modular structure of SnowClim would make this relatively straightforward.

Given the multifaceted importance of snow and ongoing snowpack changes due to climate change, there is a need for models that can accurately and efficiently simulate snow to generate spatially extensive, high-resolution datasets to meet the diverse requirements of different applications. We anticipate that the SnowClim model and data will be powerful tools for researchers

and managers across a range of disciplines including ecology and wildlife biology, recreation, transportation, hazard planning, and glacier and hydrologic modeling.

## 5 Data availability

Climate forcing data and modeled snow variables were aggregated to monthly and annual climatologies for each time period to create the SnowClim dataset (Table 5). These data are available from

https://www.hydroshare.org/resource/acc4f39ad6924a78811750043d59e5d0/ (Lute et al., 2021).



## 6 Code availability

The code for the SnowClim model is available from https://www.hydroshare.org/resource/acc4f39ad6924a78811750043d59e5d0/ (Lute et al., 2021) under the Creative Commons Attribution CC BY. The model can be run using MATLAB Online through HydroShare.


## Author Contributions

ACL developed the model code with input from JA and TL. ACL developed the downscaling code with input from JA. ACL performed the downscaling, calibration, and snow model simulations. ACL prepared the manuscript with contributions from JA and TL.


## Competing Interests

The authors declare that they have no conflict of interest.

## Acknowledgements

Support for A.C. Lute was provided by the National Science Foundation (NSF) IGERT Program (award 1249400) and a
Hydroinformatics Innovation Fellowship provided by CUAHSI with support from the NSF Cooperative Agreement No. EAR-1849458.
This research was conducted on the homelands of the Nimiipuu (Nez Perce), at the University of Idaho in Moscow, Idaho. The University of Idaho is a land grant university whose endowment was partially funded by more than 87,000 acres of land acquired through the Morrill Act. The United States government used treaties and seizures to obtain land from the Shoshone-
Bannocks, Nimiipuu, and Schitsu'umsh (Couer d'Alene) tribes. The income generated from this land, benefitting the university, is 372 times the amount paid to the tribes (Lee, 2020). Additionally, data used in this research was collected on land significant to many Indigenous communities across the western United States. The authors recognize, pay respect, and extend gratitude to the Indigenous communities that live upon, hold sacred, and care for the lands reflected in this research.

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




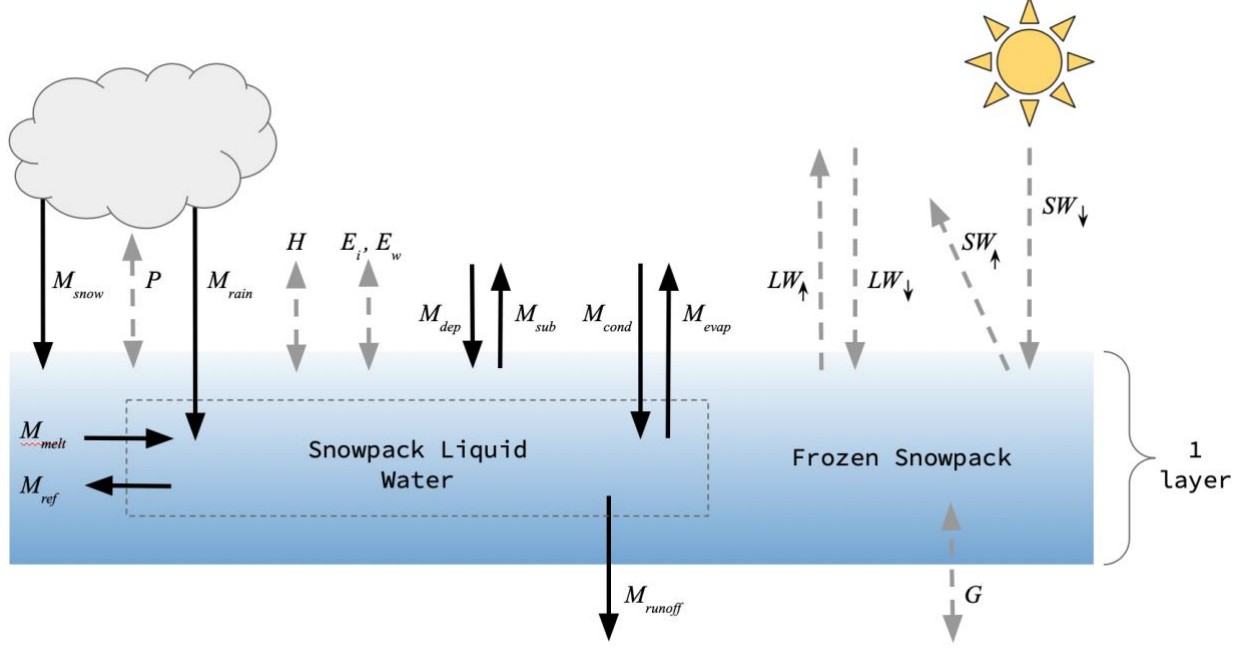

**Figure 1: Snow model conceptual diagram. Solid black arrows indicate mass fluxes, dashed grey arrows indicate energy fluxes. Fluxes are described in the text.**


| Forcing data | Abbreviation |
|---|---|
| Downward shortwave radiation flux at the surface | $SW_\downarrow$ |
| Downward longwave radiation flux at the surface | $LW_\downarrow$ |
| Air Temperature | $T_a$ |
| Dewpoint Temperature | $T_d$ |
| Precipitation | $P$ |
| Relative Humidity | $RH$ |
| Specific Humidity | $Q_a$ |
| Wind speed | $U_a$ |
| Air pressure | $P_{air}$ |



**Table 1. Required forcing data for the snow model.**

| Parameter | Abbreviated name | Values used for calibration | Units |
|---|---|---|---|
| Albedo Algorithm | *albedo_opt* | Essery[a], Tarboton[b]* | - |
| Momentum roughness length | $z_0$ | $10^{-3}$, $10^{-4}$, $10^{-5}$* | m |
| Heat and vapor roughness length | $z_h$ | $z_0/10$* | m |
| Maximum Albedo | *albedo_max* | 0.85*, 0.90 | - |
| Maximum liquid water fraction | *lw_max* | 0.1* | - |
| Windless heat exchange coefficient | *E0* | 0, 1*, 2 | $Wm^{-2}K^{-1}$ |
| Windless heat exchange coefficient flux application | *E0_app* | 1* | - |
| Windless heat exchange coefficient stability condition | *E0_stability* | 2* | - |
| Cold content threshold at which to start energy tax | $cc_0$ | 0*, -5000, -10000 | $kJm^{-2}$ |
| Cold content range to tax | $cc_1$ | -5000, -10000*, -15000, -20000 | $kJm^{-2}$ |
| Maximum tax to apply to surface energy | *maxtax* | 0.3, 0.6, 0.9* | - |


| Surface energy flux smoothing window | *smooth_hrs* | 8, 12*, 24 | hours |
| Snow surface temperature augmentation | $T_{add}$ | 0, 1, 2* | °C |

**Table 2. Parameters, their abbreviated names, the parameter values used in calibration, and their units. Parameter values with an**
**\* indicate values chosen for the full model run by calibration at SNOTEL sites. Additional parameter options, including the VIC model albedo option, were evaluated in preliminary work but were excluded from the full calibration due to consistently poor performance. [a]Essery et al., (2013); [b]Tarboton and Luce, (1996)**

| WRF data | Abbreviation |
| --- | --- |
| Downward shortwave radiation flux at the surface | $SW_↓$ |
| Downward longwave radiation flux at the surface | $LW_↓$ |
| Mean Air Temperature | $T_a$ |
| Precipitation | $P$ |
| Wind speed | $U_a$ |
| Air pressure | $P_{air}$ |
| Water vapor mixing ratio (kg/kg) | $Q$ |

**Table 3. WRF data used to derive forcing data for the snow model.**





**Figure 2. Performance metrics for an hourly model run with the selected parameterization.**

| Metric | Coefficient of determination ($R^2$) |
| --- | --- |
| Maximum SWE | 0.91 |
| Day of maximum SWE | 0.77 |
| Snow duration | 0.85 |
| Number of snow cover days | 0.89 |





| Day of snow melt out | 0.82 |
|---|---|

**Table 4. Spatial correlations ($R^2$) between observations at SNOTEL sites and SnowClim simulations for various snow metrics over the model calibration period 1 October 2000 - 30 September 2013.**


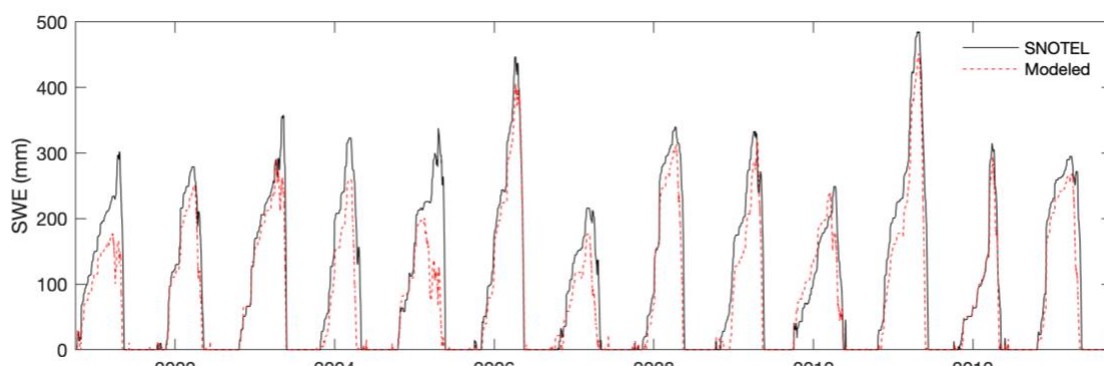

**Figure 3. Time series of observed and modeled SWE at the Hilts Creek, Idaho SNOTEL site. Out of all 170 SNOTEL sites, errors at this site were closest to the all-station median errors reported in the text.**




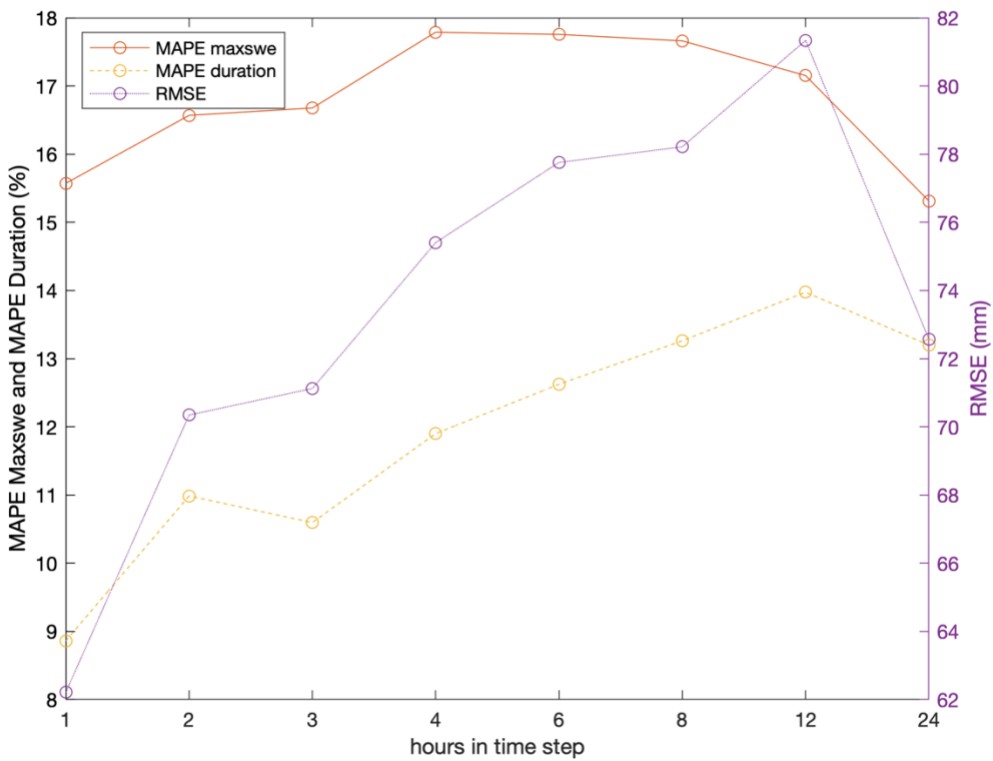

**Figure 4. Snow model performance for different time steps using the parameter set selected in calibration of the hourly model. Points represent median values across 170 SNOTEL sites.**

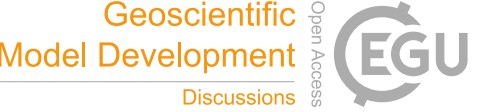




**Figure 5. a) Historical and b) future maxswe (mm), c) historical and d) future snow duration (days). Historical values are averages over the period 2000-2013. Future values represent averages during the period 2071-2100 under RCP 8.5. In a) and b), white land areas denote areas that had less than 50 mm maxswe. In c) and d), white land areas denote areas where snow duration was 0. Note the non-linear color scale in panels a) and b).**






**Figure 6. a) Absolute and b) percent change in maxswe between historical and future periods. c) Absolute and d) percent change in snow duration between historical and future periods. In d), small boxes in Utah and Colorado indicate the regions highlighted in Fig. 7 and Fig. 8, respectively.**




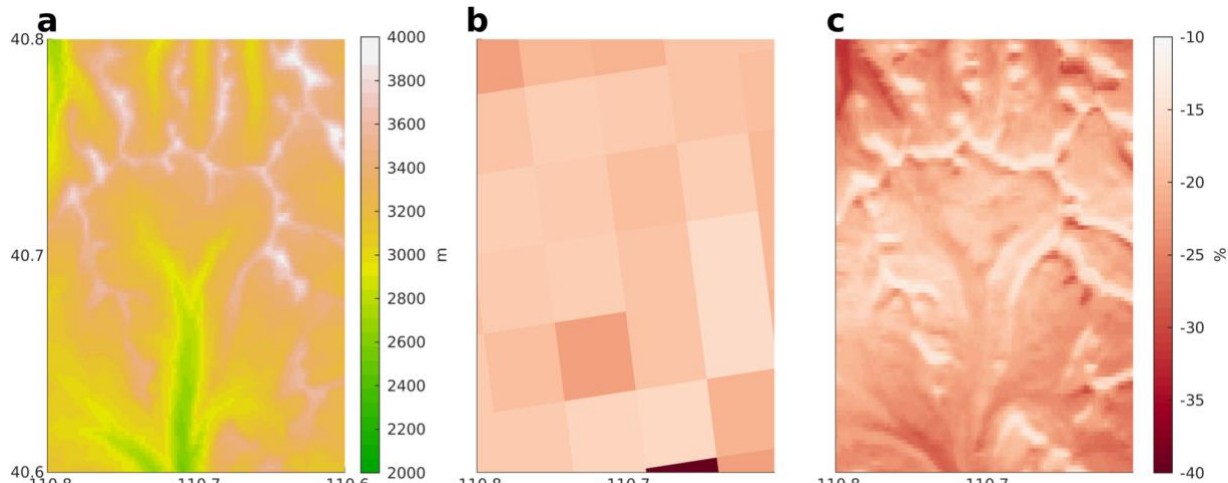

**Figure 7. Example of simulations of changing maxswe for a portion of the Uinta Mountains, Utah (location is marked in Fig. 6). The elevation (m) of the domain is shown in a). The percent change (%) in maxswe between historical and late 21st century periods as simulated by a 4 km WRF product (Rasmussen and Liu, 2017) is shown in b) and the same metric but from the SnowClim dataset is shown in c).**


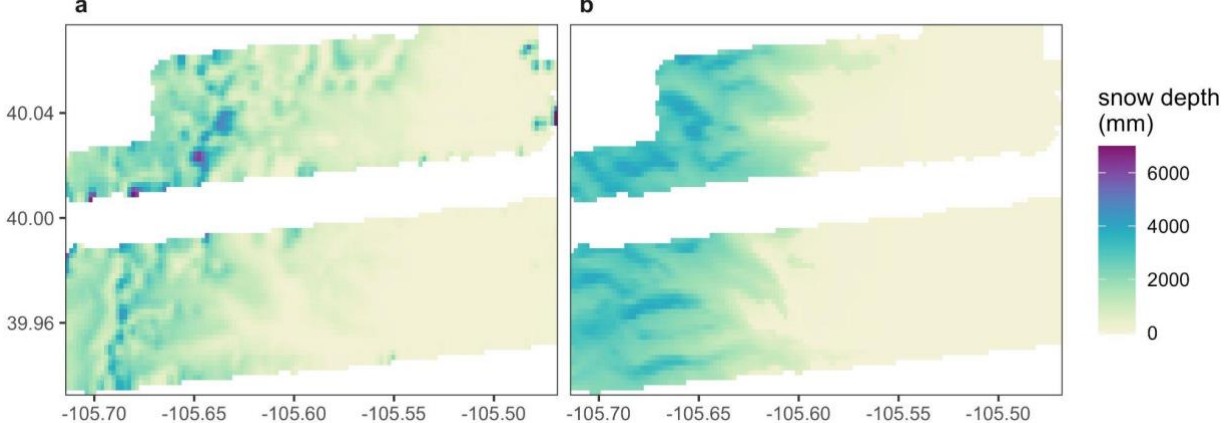

**Figure 8. Snow depth on 20 May 2010 in the Boulder Creek Watershed area indicated in Fig. 6 from a) LiDAR (described in Harpold et al., 2014) and b) SnowClim.**


| Climate Variables |
|---|
| Monthly temperature (min, max, and mean) |
| Monthly precipitation |
| Monthly solar radiation |





| |
| --- |
| Monthly dewpoint temperature |
| Annual number of freeze/thaw cycles |
| Snow Variables |
| Monthly SWE |
| Monthly snow depth |
| Monthly snow cover days |
| Monthly snowfall |
| Annual size and date of maximum SWE |
| Annual size and date of largest snowfall event |
| Annual snow duration |
| Date of first and last snow |
| Number of days without snow between first and last snow |

**Table 5. Summary climate and snow variables included in the SnowClim dataset. Summary variables are available for pre-industrial, historical, and future time periods.**





**Appendix A**

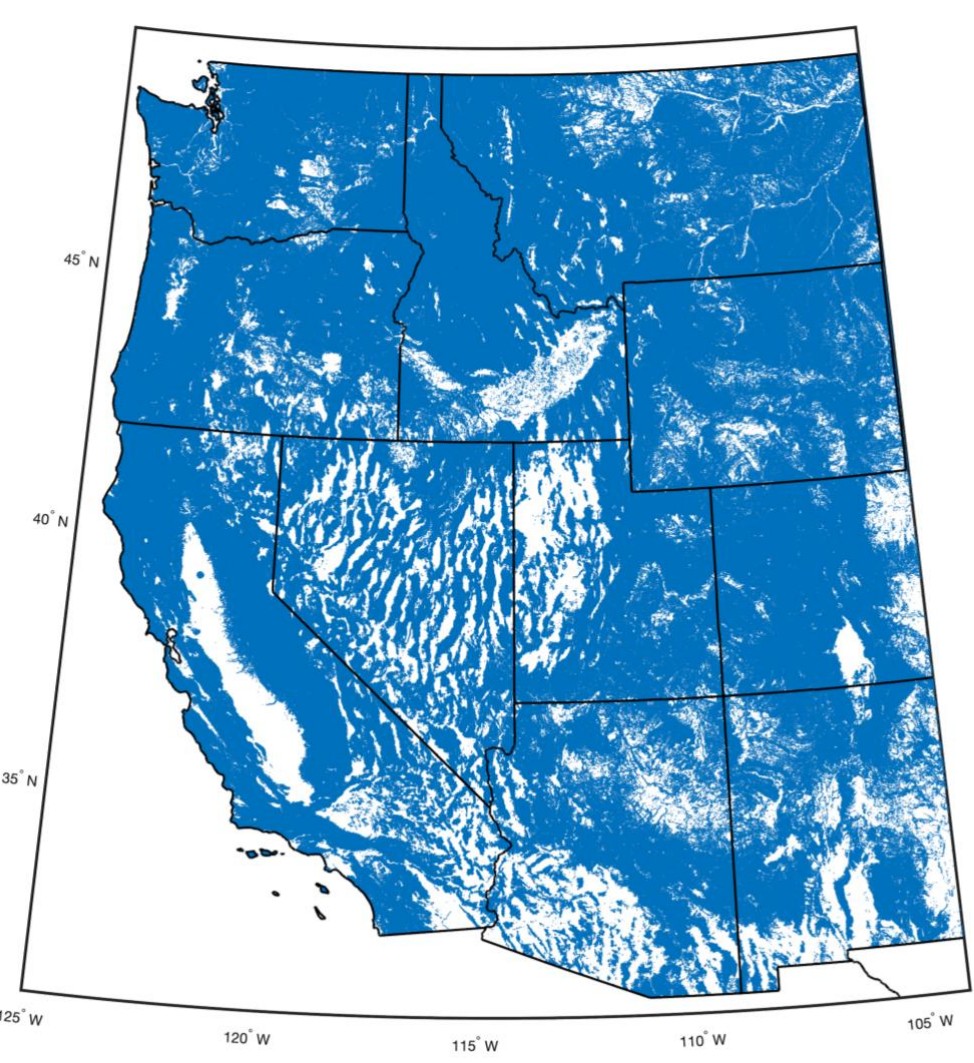


**Figure A1. Map of modeling domain with locations modeled at 210m spatial resolution in blue.**





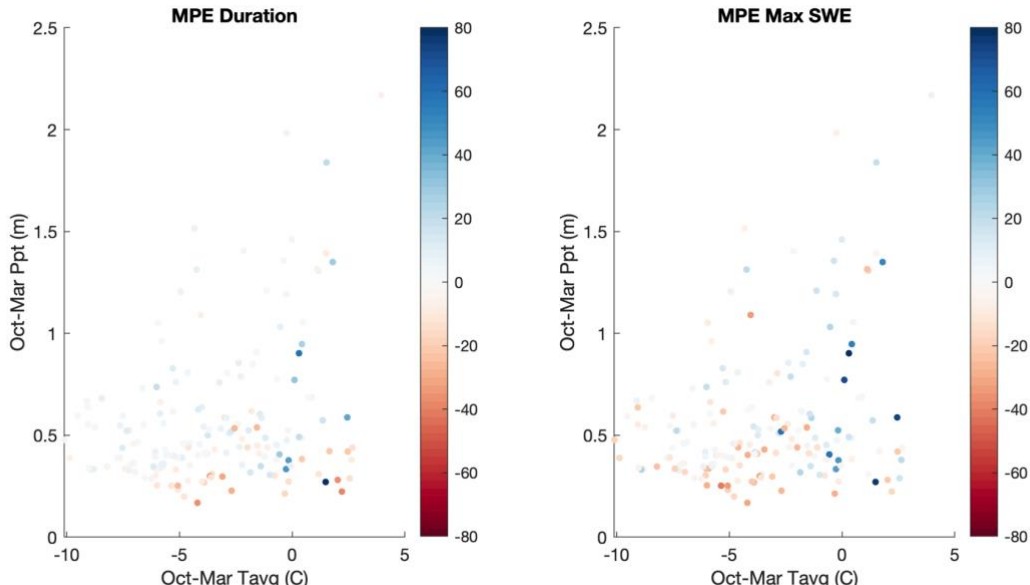

**Figure A2. Performance of best hourly model at SNOTEL sites in temperature-precipitation space. Each point represents a SNOTEL**
**site.**



**Figure A3. Parameter sensitivity of hourly model performance.**



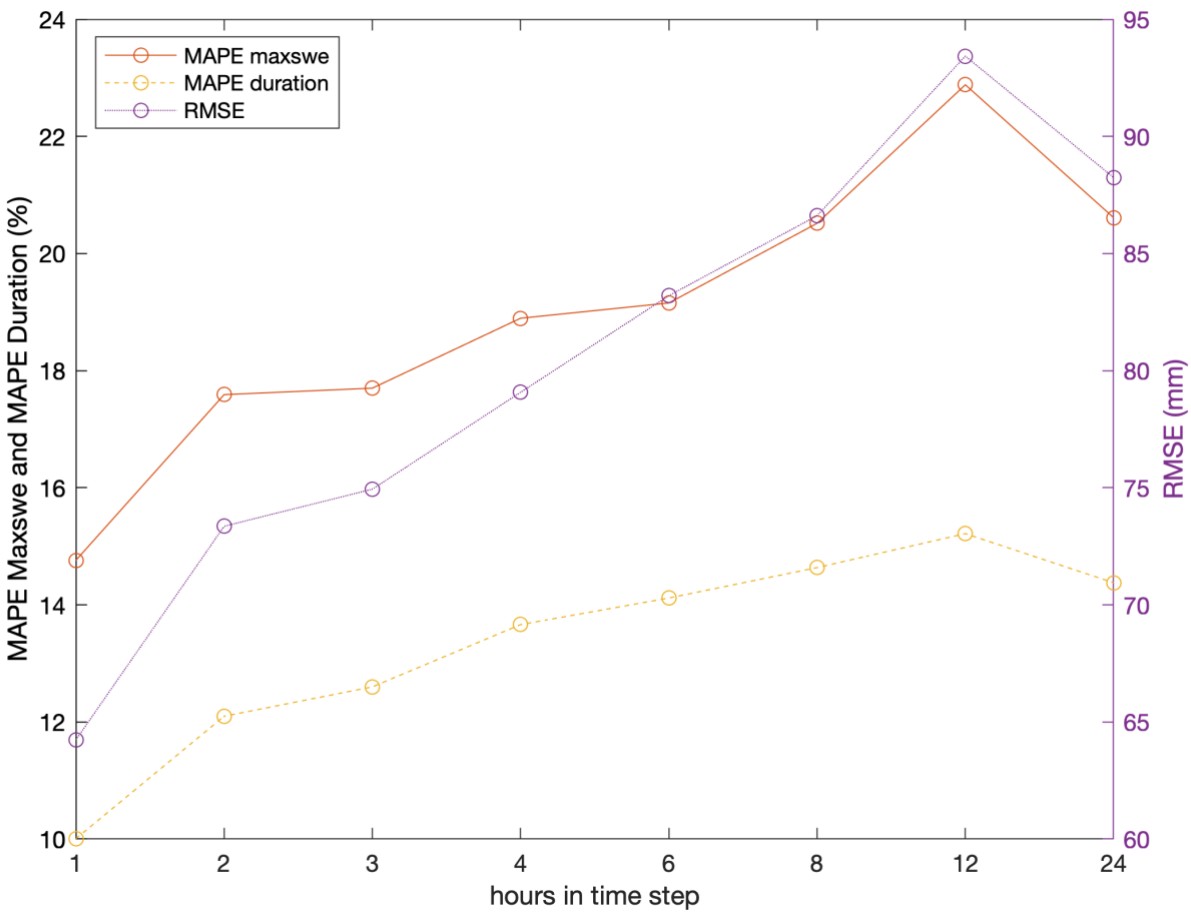


**Figure A4. Performance of snow model without shallow snow correction for different time steps using the parameter set selected in calibration of the hourly model with shallow snow correction. Points represent median values across 170 SNOTEL sites.**

| 4-Digit HUC | HUC Name | Total Area | Modeled Area | Historical Maxswe | Future Maxswe | Historical Snow Duration | Future Snow Duration | Absolute Change Maxswe | Percent Change Maxswe | Absolute Change Snow Duration | Percent Change Snow Duration |
|---|---|---|---|---|---|---|---|---|---|---|---|
| 901 | Souris | 26321 | 1100 | 26 | 15 | 115 | 63 | -11 | -42 | -52 | -45 |
| 904 | Saskatchewan River | 17604 | 1748 | 617 | 434 | 237 | 183 | -183 | -30 | -54 | -23 |
| 1002 | Missouri Headwaters | 36350 | 36350 | 226 | 144 | 199 | 137 | -82 | -36 | -62 | -31 |
| 1003 | Missouri-Marias | 51423 | 51423 | 79 | 49 | 144 | 93 | -30 | -38 | -51 | -36 |
| 1004 | Missouri-Musselshell | 60830 | 60830 | 32 | 18 | 115 | 63 | -15 | -46 | -52 | -45 |
| 1005 | Milk | 59844 | 38732 | 22 | 13 | 96 | 52 | -9 | -39 | -45 | -47 |





| 1006 | Missouri-Poplar | 34275 | 28125 | 19 | 11 | 97 | 50 | -8 | -41 | -47 | -48 |
|------|------|------|------|------|------|------|------|------|------|------|------|
| 1007 | Upper Yellowstone | 37453 | 37453 | 196 | 140 | 175 | 118 | -56 | -29 | -56 | -32 |
| 1008 | Big Horn | 59250 | 59250 | 93 | 60 | 153 | 100 | -33 | -35 | -53 | -34 |
| 1009 | Powder-Tongue | 48695 | 48695 | 29 | 15 | 127 | 66 | -14 | -49 | -61 | -48 |
| 1010 | Lower Yellowstone | 35982 | 35982 | 17 | 8 | 96 | 47 | -9 | -51 | -49 | -51 |
| 1011 | Missouri-Little Missouri | 41312 | 26872 | 22 | 9 | 100 | 46 | -12 | -57 | -54 | -54 |
| 1012 | Cheyenne | 61764 | 44105 | 32 | 16 | 123 | 66 | -16 | -50 | -57 | -47 |
| 1013 | Missouri-Oahe | 39533 | 6792 | 18 | 8 | 90 | 41 | -10 | -54 | -49 | -54 |
| 1014 | Missouri-White | 17839 | 1738 | 21 | 11 | 102 | 48 | -10 | -48 | -53 | -52 |
| 1015 | Niobrara | 17280 | 4959 | 19 | 10 | 121 | 42 | -9 | -48 | -80 | -66 |
| 1018 | North Platte | 78925 | 68121 | 80 | 53 | 153 | 96 | -27 | -33 | -57 | -37 |
| 1019 | South Platte | 61585 | 54444 | 49 | 30 | 130 | 68 | -19 | -38 | -62 | -48 |
| 1025 | Republican | 34705 | 2048 | 14 | 7 | 75 | 25 | -7 | -50 | -49 | -66 |
| 1102 | Upper Arkansas | 64555 | 50655 | 42 | 28 | 101 | 52 | -15 | -34 | -49 | -49 |
| 1104 | Upper Cimarron | 20693 | 3041 | 22 | 14 | 67 | 26 | -8 | -36 | -41 | -61 |
| 1108 | Upper Canadian | 32307 | 31937 | 25 | 15 | 71 | 32 | -10 | -41 | -39 | -55 |
| 1109 | Lower Canadian | 23149 | 4348 | 15 | 9 | 45 | 19 | -6 | -38 | -26 | -58 |
| 1110 | North Canadian | 20259 | 1518 | 20 | 13 | 48 | 25 | -7 | -33 | -23 | -48 |
| 1112 | Red Headwaters | 20120 | 1170 | 14 | 8 | 37 | 14 | -6 | -42 | -24 | -63 |
| 1205 | Brazos Headwaters | 33307 | 5234 | 9 | 3 | 23 | 3 | -6 | -66 | -20 | -88 |
| 1208 | Upper Colorado | 39084 | 5265 | 8 | 2 | 13 | 2 | -6 | -74 | -11 | -87 |
| 1301 | Rio Grande Headwaters | 19715 | 19715 | 166 | 116 | 162 | 109 | -50 | -30 | -52 | -32 |
| 1302 | Rio Grande-Elephant Butte | 70248 | 70248 | 50 | 26 | 73 | 33 | -24 | -47 | -40 | -55 |
| 1303 | Rio Grande-Mimbres | 56317 | 28836 | 7 | 1 | 11 | 2 | -6 | -81 | -10 | -87 |
| 1304 | Rio Grande-Amistad | 73020 | 4701 | 4 | 0 | 4 | 0 | -4 | -91 | -4 | -99 |
| 1305 | Rio Grande Closed Basins | 45513 | 38932 | 11 | 4 | 26 | 5 | -7 | -68 | -20 | -79 |
| 1306 | Upper Pecos | 60947 | 60947 | 14 | 7 | 29 | 8 | -7 | -52 | -21 | -74 |
| 1307 | Lower Pecos | 53426 | 14078 | 3 | 1 | 4 | 0 | -3 | -75 | -4 | -99 |
| 1401 | Colorado Headwaters | 25480 | 25480 | 279 | 190 | 196 | 143 | -89 | -32 | -53 | -27 |
| 1402 | Gunnison | 20791 | 20791 | 244 | 169 | 182 | 130 | -75 | -31 | -52 | -28 |



| 1403 | Upper Colorado-Dolores | 21662 | 21662 | 117 | 64 | 124 | 70 | -53 | -45 | -54 | -43 |
|------|------------------------|-------|-------|-----|-----|-----|-----|------|-----|-----|-----|
| 1404 | Great Divide-Upper Green | 53758 | 53758 | 89 | 59 | 151 | 92 | -30 | -34 | -60 | -39 |
| 1405 | White-Yampa | 34342 | 34342 | 183 | 113 | 162 | 111 | -69 | -38 | -51 | -32 |
| 1406 | Lower Green | 37701 | 37701 | 123 | 75 | 129 | 78 | -47 | -38 | -52 | -40 |
| 1407 | Upper Colorado-Dirty Devil | 35265 | 35265 | 46 | 22 | 76 | 30 | -25 | -53 | -46 | -61 |
| 1408 | San Juan | 64570 | 64570 | 79 | 44 | 86 | 37 | -36 | -45 | -49 | -57 |
| 1501 | Lower Colorado-Lake Mead | 78401 | 78401 | 35 | 12 | 56 | 18 | -24 | -67 | -38 | -68 |
| 1502 | Little Colorado | 70078 | 70078 | 25 | 7 | 65 | 16 | -18 | -72 | -49 | -75 |
| 1503 | Lower Colorado | 53742 | 44549 | 4 | 1 | 7 | 0 | -3 | -79 | -7 | -96 |
| 1504 | Upper Gila | 39347 | 39347 | 22 | 5 | 37 | 10 | -17 | -77 | -27 | -74 |
| 1505 | Middle Gila | 46573 | 43759 | 4 | 0 | 5 | 0 | -3 | -88 | -5 | -94 |
| 1506 | Salt | 34899 | 34899 | 53 | 13 | 63 | 19 | -40 | -75 | -44 | -70 |
| 1507 | Lower Gila | 39046 | 39046 | 2 | 0 | 4 | 0 | -2 | -84 | -3 | -95 |
| 1508 | Sonora | 62266 | 12927 | 2 | 0 | 3 | 0 | -2 | -83 | -2 | -90 |
| 1601 | Bear | 19464 | 19464 | 226 | 120 | 177 | 117 | -106 | -47 | -59 | -34 |
| 1602 | Great Salt Lake | 74295 | 74295 | 96 | 43 | 114 | 54 | -53 | -55 | -60 | -53 |
| 1603 | Escalante Desert-Sevier Lake | 42670 | 42670 | 93 | 45 | 129 | 70 | -48 | -51 | -59 | -46 |
| 1604 | Black Rock Desert-Humboldt | 74178 | 74178 | 67 | 25 | 122 | 60 | -42 | -63 | -63 | -51 |
| 1605 | Central Lahontan | 32838 | 32838 | 96 | 45 | 95 | 45 | -51 | -53 | -50 | -52 |
| 1606 | Central Nevada Desert Basins | 123606 | 123606 | 41 | 16 | 95 | 43 | -24 | -59 | -52 | -54 |
| 1701 | Kootenai-Pend Oreille-Spokane | 134753 | 94016 | 400 | 202 | 189 | 125 | -198 | -50 | -64 | -34 |
| 1702 | Upper Columbia | 102909 | 57668 | 213 | 103 | 125 | 64 | -111 | -52 | -61 | -49 |
| 1703 | Yakima | 15928 | 15928 | 322 | 126 | 128 | 67 | -196 | -61 | -62 | -48 |
| 1704 | Upper Snake | 92909 | 92909 | 209 | 129 | 160 | 99 | -80 | -38 | -61 | -38 |
| 1705 | Middle Snake | 95797 | 95797 | 169 | 78 | 130 | 66 | -91 | -54 | -63 | -49 |
| 1706 | Lower Snake | 90765 | 90765 | 339 | 178 | 176 | 111 | -161 | -48 | -65 | -37 |
| 1707 | Middle Columbia | 77449 | 77449 | 150 | 45 | 108 | 45 | -105 | -70 | -63 | -58 |
| 1708 | Lower Columbia | 16121 | 15815 | 318 | 74 | 112 | 41 | -245 | -77 | -71 | -63 |

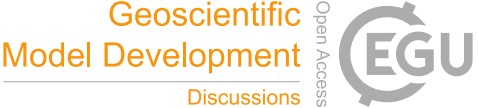

| | | | | | | | | | | |
|---|---|---|---|---|---|---|---|---|---|---|
| 1709 | Willamette | 29699 | 29699 | 207 | 47 | 87 | 31 | -160 | -77 | -56 | -64 |
| 1710 | Oregon-Washington Coastal | 65238 | 59592 | 129 | 25 | 80 | 19 | -104 | -80 | -61 | -76 |
| 1711 | Puget Sound | 52957 | 36404 | 490 | 155 | 126 | 60 | -334 | -68 | -65 | -52 |
| 1712 | Oregon Closed Basins | 45143 | 45143 | 78 | 26 | 125 | 54 | -51 | -66 | -71 | -57 |
| 1801 | Klamath-Northern California Coastal | 67762 | 64619 | 195 | 51 | 108 | 47 | -143 | -74 | -62 | -57 |
| 1802 | Sacramento | 72013 | 72013 | 195 | 58 | 93 | 46 | -137 | -70 | -47 | -50 |
| 1803 | Tulare-Buena Vista Lakes | 42498 | 42498 | 107 | 59 | 52 | 32 | -48 | -45 | -21 | -40 |
| 1804 | San Joaquin | 40984 | 40984 | 196 | 107 | 68 | 43 | -88 | -45 | -25 | -37 |
| 1805 | San Francisco Bay | 13910 | 11448 | 1 | 0 | 1 | 0 | -1 | -97 | -1 | -97 |
| 1806 | Central California Coastal | 34287 | 29377 | 5 | 0 | 4 | 0 | -4 | -90 | -4 | -93 |
| 1807 | Southern California Coastal | 35865 | 28785 | 18 | 3 | 16 | 3 | -15 | -84 | -13 | -82 |
| 1808 | North Lahontan | 11791 | 11791 | 101 | 33 | 138 | 72 | -68 | -67 | -66 | -48 |
| 1809 | Northern Mojave-Mono Lake | 73268 | 73268 | 21 | 10 | 24 | 11 | -11 | -51 | -14 | -56 |
| 1810 | Southern Mojave-Salton Sea | 44247 | 41522 | 3 | 1 | 3 | 0 | -3 | -84 | -2 | -84 |
| Total | Western US | 3794897 | 3100511 | 111 | 53 | 101 | 54 | -58 | -52 | -47 | -47 |

**Table A1. Summary of snow metrics by 4-digit HUC. Total and modeled areas are in km2, maxswe has units of mm, and snow duration has units of days. Values are averaged over all modeled grid cells within each HUC. Total snow metric values, in the last row, are averaged across all grid cells across the western U.S. modeling domain and total changes are computed from these total snow metric values.**