# Peer review of "SnowClim v1.0: High-resolution snow model and data for the western United States"

_Geoscientific Model Development, 2021_

## Author Comment (AC4)

**Referee comments are in black, author responses are in green.**

Lute et al. present simulations of snow over a large domain for reasonably long periods and at high spatial resolution. To achieve this, they have made some quite severe approximations to limit computational expense. Neglecting forest effects seems like a limitation for water resource applications in the western US. Simple canopy models, and snow models with a few layers to allow a more physical representation, need not be very expensive. Has code optimization to offset the cost of improved model physics been considered? MATLAB is convenient for parallel computation, but otherwise might not be the best language for this. Having said that, the authors have adopted a particular development strategy and reporting the results is worthwhile.

We appreciate the thoughtful and thorough comments of the referee.

The referee is correct that in the interest of spatial resolution, and temporal and spatial extent, some simplifications were made – which we tried to be transparent about. That said, the SnowClim model presents a much more physically based approach than the temperature-index modeling approaches that are often used when these levels of extent and resolution are required (e.g. SNOW-17, Anderson, 2006; Kraaijenbrink et al., 2021).

Substantial effort was put into code optimization. Model calibration and the model runs for the Western United States were done using parallel computation on high performance computers. A variable spatial resolution was adopted in order to enable the high spatial resolution and extent. As discussed in the manuscript, the barriers to inclusion of more comprehensive process representations include not only computational expense, but also appropriate data. In the case of forest effects, data to inform a vegetation module for the pre-industrial, recent historical, and future time periods is not available. In addition, there is significant uncertainty in future vegetation change stemming from disturbance regimes and species composition pathways which would add complexity and uncertainty to the model. That said, neglecting forest effects is certainly a limitation of the model and represents one of the top areas for future development that could be added to the modeling approach presented. As it is, the SnowClim data represents a first approximation of snow in vegetated landscapes with demonstrated accurate performance at open locations. These points have been incorporated in the revised manuscript.

I was able to access the code and run the MATLAB Online example easily (a quickstart guide might have saved me a couple of minutes).

I suggest some corrections and clarifications, identified by line number:

24

There are too many to list, but the abstract should give some idea of what kind of "snow metrics" are offered.

We appreciate this suggestion and have added some metrics to line 26.

52

Should note that this is earlier but slower snowmelt

Yes, good point. We have modified the sentence to this effect

106

Equation (1) is not the surface energy balance because G is not a surface flux. I would describe it as the energy balance of the snowpack.

Thank you for raising this point, we agree. We'll use 'snowcover energy balance' instead of 'surface energy balance'. We have modified the text to reflect this.

127

Note that separate visible and near-infrared radiation fluxes are not supplied as inputs; the albedo is simply averaged between these bands. The illumination angle dependence should only be applied to the direct-beam component of incoming radiation.

You are correct that separate radiation fluxes are not supplied to the albedo calculation, we have modified the text to reflect this. And yes, the albedo is the average of the near infrared and visible band albedos.

Regarding the illumination angle, the model does not differentiate the direct-beam and diffuse radiation. Instead, the inclination angle adjustment is applied to global radiation. While this is not technically correct, the parameterization in UEB (the source of this albedo function) is calibrated for global radiation. We also note that this simplification increases computational efficiency and recognizes that diffuse radiation provides a minor contribution to most shortwave radiation driven snowmelt.

133

Equation (3) is incorrect. If the emissivity is not 1, the upward longwave radiation includes a reflected fraction of the downward radiation (Kirchoff's law).

Thank you for catching this. We have corrected the equation, recalibrated the model, and recompleted the western US runs with the new equation and parameters.

179

Constant G is not a very common (or realistic) feature of modern models. Etchevers et al. (2002) included an experiment with fixed G as a sensitivity study, but did not report the results.

While constant G is not very common in modern physics based snow models, G is entirely neglected (or you might say subsumed) in empirical (e.g. temperature-index)

snow models. We recognize the referee's point however that the manuscript text might be misleading in that it might suggest that the use of a constant value for G in physics based snow models is more common than it actually is. We have added a sentence to the manuscript in section 2.2.5 to clarify this point and more accurately reflect common practice.

Given unlimited computational capacity and time, incorporating a more thorough treatment of the ground heat flux would be ideal. However, this would require dynamic soil temperatures, which can vary significantly in complex terrain and require the addition of a soil temperature module. The ground heat flux also depends on soil properties including texture, water content, frozen/unfrozen status, as well as litter depth, characteristics, and water content. The benefits of modeling a dynamic ground heat flux likely do not justify the added complexity and uncertainty it would require since in most mountain areas the ground heat flux is small compared to other fluxes (Marks and Dozier, 1992; DeWalle and Rango, 2008).

Given that the goals of the SnowClim model were to incorporate the most important elements of physics based models and to include empirical components to enable greater computational efficiency, we feel that the use of a constant value for G is entirely appropriate. A brief statement to this effect is included in the revised manuscript.

189

Iterative solution of the surface energy balance to find snow surface temperature is not the only possibility. Best et al. (2011, https://gmd.copernicus.org/articles/4/677/2011/), for example, linearizes the surface energy balance equation to find a surface temperature solution without the expense of iteration. In either an iterative or linearised solution, the aim is to find a surface temperature that is consistent with the surface energy balance. I don't see how this can be achieved with the pragmatic but ad hoc approach in SnowClim.

We thank the reviewer for highlighting this additional option. We think the linearization approach could be worth considering in a future version of the SnowClim model, but would require thorough evaluation in the context of the existing model. Furthermore, the point referenced in the manuscript describes options for solving for snow surface temperature in single layer models, however it appears that the linearization approach in Best et al., (2011) is only applied to the multilayer model formulation. We note that the single layer UEB model uses a combination of linearization and iteration to solve for snow surface temperature. We have clarified the text to indicate that iterative and linearization approaches are often used, instead of just noting the iterative option.

281

How were the 210 m and 1050 m resolutions selected? How many points are there in this domain?

The spatial resolutions were selected based on several considerations. For ease of implementation, we chose spatial resolutions that were multiples of 30 m, the spatial resolution of the digital elevation model we used. Beyond that, we selected 1050 m based on the assumption that snow conditions could be relatively uniform across this extent given minimal contrast in elevation and solar radiation (our conditions for a location to be modeled at coarse resolution). At the higher end, we selected a spatial resolution that was not so high as to make our spatial and temporal extent goals unachievable.

We modeled 920,605 points at 1050 m resolution and 64,310,454 points at 210 m resolution. We've noted this in the revised manuscript.

330

SNOTEL sites may not be representative of larger areas. Could this calibration skew the model performance?
shttps://agupubs.onlinelibrary.wiley.com/doi/10.1029/2005WR004229

We agree that SNOTEL sites may not be representative of larger areas (just as any point on the landscape may not be representative of the larger area). SNOTEL sites tend to be located in forest clearings which can have distinctly different wind and radiational characteristics compared to closed canopy forests. If there was a consistent difference across sites between SNOTEL SWE and broader area SWE then calibrating at SNOTEL sites could skew the model performance across broader domains.

Essentially any existing observational network could be less than a perfect representation of the broader area and potentially skew the model results. The SNOTEL sites provide the best available calibration dataset in the study region due to length of records, relatively consistent observational equipment and methods across sites, and the breadth of geographic and climatic conditions that they cover. We also note in the manuscript that the SnowClim model can be calibrated to other observational datasets, however we calibrate to SNOTEL in this case for the reasons aforementioned. We've added some text in the calibration methods section (3.3.1) to acknowledge the concern about representativeness and indicate why we chose to use the SNOTEL network for calibration.

362

Any ideas why model performance would improve from 12 to 24 hour timesteps?

We suspect that the improvement in model performance between 12 and 24 hour timesteps for the time aggregation windows shown in Figure 4 is related to how well the aggregation windows capture the diurnal cycle of energy fluxes. For the 12 hour aggregation starting at 0 GMT (shown in Figure 4), solar radiation oscillates between 0 or a low value and a moderately high value, with nothing in between. In contrast, the 24 hour aggregation window starting at 0 GMT (shown in Figure 4) has moderate solar radiation values in every time step.

We found that if we started the aggregation windows a few hours later (e.g. at 3 GMT or later), then the model performance degraded continuously with coarsening temporal resolution. These aggregation windows allow a more even distribution of solar radiation values across time steps. A note regarding this has been added to the revised manuscript.

369

There is no evaluation of the large-scale snow simulations for the western US. MODIS snow cover extent products would have a convenient resolution for this.

We do not think a comparison with MODIS snow cover extent data would be very illuminating given that it is just a binary snow/no snow product. Recent studies (e.g., Garousi-Nejad and Tarboton 2022) have demonstrated several challenges in comparing modeled snow to the MODIS snow cover area product including factors such as forest cover that may yield incorrect estimates from MODIS. While a comparison to another large-scale dataset could be useful, we feel it is beyond the scope of this paper given the evaluations we have already provided.

Garousi-Nejad, I., & Tarboton, D. G. (2022). A comparison of National Water Model retrospective analysis snow outputs at snow telemetry sites across the Western United States. *Hydrological Processes*, 36( 1), e14469. https://doi.org/10.1002/hyp.14469

Figure 2

There are a lot of sites to balance in the calibration, but errors up to +/- 50 days in duration and +/- 50% in maximum SWE even after calibration seem large for practical applications. Other datasets used to demonstrate poorer agreement in the discussion were not calibrated to SNOTEL sites.

We agree that the errors are large in a few cases, however, out of 170 sites only 9 sites had maximum SWE errors larger than +/- 50% and only 4 sites had snow duration errors larger than +/- 50 days. Our approach was to calibrate across all SNOTEL sites at once to arrive at the best parameter set across this broad range of snow climates which could then be used to model snow across the western US. If we had calibrated to each site individually we expect the errors would have been drastically smaller, at the expense of potentially having limited utility in modeling both across spatial domains and under different climates.

We also emphasize the errors in the SnowClim dataset relative to SNOTEL sites stem from a variety of factors including uncertainties and errors in data collection at SNOTEL sites (e.g. snow bridging on the snow pillow), model structure, parameterization, and forcing data. Errors in meteorological forcing, namely precipitation, may lead to biases or errors in the resultant modeled SWE that do not reflect deficiencies in the modeling framework. It is difficult to decompose these errors, but previous studies have shown that forcing data can be one of the primary contributors to overall simulation error (Günther et al., 2019).

One of the datasets we compare to (Guan et al., 2013) was not evaluated relative to SNOTEL sites. They used a combination of modeling, remotely sensed, and ground based data to simulate SWE at 6 sites in the Sierra Nevada Mountains of California. One would think that with remotely sensed and ground observation information, their approach would yield much better results than a pure modeling approach that was calibrated across a much wider range of conditions, however it did not.

Günther, D., Marke, T., Essery, R., & Strasser, U. (2019). Uncertainties in snowpack simulations—Assessing the impact of model structure, parameter choice, and forcing data error on point-scale energy balance snow model performance. Water Resources Research, 55, 2779–2800. https://doi.org/10.1029/2018WR023403

Table 1

Given temperature, pressure and one of dewpoint temperature, relative humidity and specific humidity, the other two can be calculated – they are not all required forcing data.

Yes, good point. We have revised the table to list specific humidity, but not relative humidity or dewpoint temperature since equations for calculating the latter two are available in the SnowClim model.

Table 2

As it is abbreviated as Q, I guess that the WRF mixing ratio output is taken as the SnowClim specific humidity input (they are not the same thing, but will have nearly identical values).

Specific humidity was calculated from the water vapor mixing ratio (see lines 325-326 in the initial manuscript), they were not assumed to be equal.

Table 5

The monthly SWE and snow depth are minimum, mean and maximum.

We have clarified this in Table 5 to better reflect the available data.

---

## Author Comment (AC5)

**Referee comments are in black, author responses are in green.**

This looks like it might be a useful addition to the suite of datasets of snow simulations available in the western United States. The combination of a reasonable physical basis, validation against observations in diverse climates, and an underlying high resolution provides, at least conceptually, a reasonably satisfying sense that the estimates provided near the end in Figures 5 and 6 are sound. It seems appropriate to publish this presentation of the details of the methods used in producing these data sets. At the same time, there are improvements that could be made in the writing to provide a more constructive contribution.

The paper gives a distinct sense of critique of the existing set of snow models available. Admittedly some are related to questions about run times rather than performance, but sometimes the criticisms come across more like aspersions than measured or tested issues. In this context, I look at all the plots from SNOWMIP2 (Rutter et al. 2009), and I'm left wondering where this model would fit into the cloud of lines. The bottom line is that this is another snow model on top of an already long list, with several new approximations, assertions, and assumptions, and we are left wondering what exactly we are learning from this particular modeling exercise. The focus was on improving model speed so that larger areas could be evaluated at finer resolutions in a reasonable time, but the net gain in understanding at these larger scales is not really highlighted so much as alluded to. This is a problem because there are several products out there and in use for larger scales, and while this model conceptually critiques these products, some uncited, it only has a somewhat philosophical basis for that critique. At a basic level, the claim for superiority of this modeling approach relates to a reductionist philosophy that if we can just do our simulations at finer and finer scales, we can resolve everything. The paper ultimately bumps into the fact that as we get into finer scale simulations, we find processes that depend on adjacent areas, and the upshot of this hazard appears in line 416 – near the end of the paper – where they note that the model is "most realistic in minimally vegetated areas with relatively little snow redistribution", a rare piece of ground in the western US.

So, the question remains about what we learn from this modeling exercise. The central claim seems to be that when we have more resolution, we can resolve effects of things like aspect and solar radiation, so we have a better representation than we have had in the various existing undisclosed products. This may be true, but at the same time the paper's central argument is fundamentally recommending further reduction in element scale with addition of vegetation and redistribution effects. Is there simply an argument for just doing the grind of getting complete lidar coverage over the western US, including data that can tell us about accumulation enhancement/loss patterns, and running a hyper-resolution snow model everywhere? Is this the only way that we can ultimately get a satisfying snow data set? At what scale does a reductionist argument end? Is there any utility in considering subgrid parameterizations of the effects of solar radiation, redistribution, and forests to a grid scale that is useful … or is that just a side-track that gets swallowed up by research in computing, storage, and remote sensing? My point is that this paper – though currently unstated – takes a stand with respect to these

philosophical discussions in the literature. Being unstated, it is a strongly one sided stand being taken, and it's possible the authors actually have a more nuanced perspective and this paper is simply an exploration in one dimension.  In that case they might want to say something.

None of this detracts from the likelihood that this model product probably provides greater insights for some purposes about changes that are occurring than some existing products, but it is important to examine and state in the introduction what the central argument is and its philosophical history.  At the very least, this makes it easier for others to follow up on the conceptual advance of the paper.  In this case, the argument seems to be that some aspects of model physics can be sacrificed in order to better incorporate the effects of spatial heterogeneity in solar radiation, elevation, and temperature on snowpack over the same area, and potentially be able to display some of the heterogeneity in snowpack at finer scales for some purposes.  This is a good question, and not entirely certain in its outcome, particularly when other subgrid processes are set aside.  I don't know that this tradeoff has been assessed in this paper so much as asserted, and it leaves on the table the question of whether a subgrid paramaterization coupled with earlier products could potentially generate greater benefits in time and accuracy … at least for purposes not needing to directly display fine-scale heterogeneity.  It also leaves open questions about which particular process for heterogeneity are most critical to incorporate if following a reductionist approach.

It is worth noting here that we have seen images very similar to Figures 5 and 6 (which display fairly coarse features) in multiple recent publications that are not necessarily mentioned or cited (including by the first author).  At some level, though, these earlier coarser resolution papers must be doing something somewhat correctly to get essentially the same images.  The abstract and introduction begin by pointing out that there are no existing data sets that are "based on physical principles, simulated at high spatial resolution, and cover large geographic domains", but they do not reflect on applications of these existing somewhat similar data sets, and clearly define "physical principles" or "high resolution" and explain for what purposes they are valuable.  These definitions probably relate to the purposes of the simulation and the scale/resolution of the display.  It is very likely that some of these existing products are adequate for many tasks, and rather than suggesting a position that this new data set makes all preceding simulations obsolete, perhaps discussing new applications the higher resolution data allow or note specific improvements they can demonstrate.  This is partially addressed in Figure 7 and references, but only for one specific application where details of heterogeneity are important, although arguably could be focused on very small areas where redistribution may be important.  On the net, if this data set is intended as replacement for the other data sets, it would be more polite to acknowledge them and their utility in the conversation so far rather than to ignore them.  If in no other way, they could be acknowledged for their corroboration of images like Figures 5 and 6.

There is a similar set of discussions about philosophical subjects with respect to the framing set up in lines 55-65.  There is a significant discussion in the hydrology literature at the moment, and spilling into geomorphology and snow as well, about the utility of

machine learning approaches.  Some argue that these are not "physically based", but others note that the calibration needs of some models renders them effectively as non-physically based as some of the machine learning approaches.  Quite a bit of work has been done in snow with ML and it is currently an area of active research.  I'd recommend the authors take a look at a brief paper by Fleming et al 2021 to see how aspects of that framing might be unhelpful to furthering the goals of their work.  I'm certainly in agreement that we all need to understand 1) the physics of the real world and 2) how our models represent that physics, but there are sound arguments that a priori assertions about the details of the physics in model formulation that could detract from our learning about the physics of the real world.  There are others who deliver that argument better than I, and hopefully after looking at some of those you might find a better way to describe the meaning of "physics-based" in the paper.

The above points are for general framing and are important to address so that the contributions of this specific modeling exercise can be better placed among much that has occurred before and much going on concurrently.

We thank the referee for their very thorough and thoughtful comments on the manuscript. In particular, we appreciate the time taken by the referee to highlight how the manuscript can better contextualize this contribution in the light of past and ongoing developments in snow modeling and the abundance of existing datasets that have tremendous value. It was certainly not our intention to discount existing snow data products or approaches, however we can see how the limited space devoted to this topic in the manuscript and the focus on our new approach could give that impression. To address this, we have better contextualized the current work by noting in the introduction the limitations of our approach and applications for which existing approaches would be better suited. We have also added a sentence in the section presenting figures 5 and 6 that acknowledges the broad similarity of these results to previous work. We have also added text in the Conclusion to further highlight the strengths of our approach and what can be gained from it (a better understanding of the effects of elevation and aspect on climate change sensitivity of snow, highlighting potential snow refugia). We believe the discussion of drawbacks of our approach in the Conclusion is sufficiently thorough, however we have more clearly indicated that some of these drawbacks are overcome by existing approaches/products.

One of the objections of the referee was with the lack of definitions for the terms physics based and high spatial resolution. We have clarified these terms in the revised manuscript (section 1, paragraph 3), defining physics based as incorporating process based equations for radiative and turbulent energy exchanges and defining high spatial resolution as $< 1 \text{ km}^2$. We acknowledge that the lines between empirical methods (such as temperature index approaches), machine learning (ML) and artificial intelligence (AI) approaches, and traditional physics based modeling are blurry and recognize that the former two categories can incorporate physics in some form while the latter can be empirical to some extent (due to numerous non-physical calibration parameters). Indeed the latter may be more accurately called process based modeling than physics based modeling. In light of these nuances, we have largely changed the text to use the term process-based instead of physics-based. We use process-based to recognize the inclusion of essential physics equations in the model (e.g. equations 2, 3, and to some

extent the equations for turbulent fluxes), however we emphasize that the SnowClim model takes a hybrid approach- blending elements that are typically included in physics-based models with more empirical elements. If anything, the SnowClim model further emphasizes the lack of strict boundaries between what is a process-based (typically referred to as physics-based) model and what is an empirical model. We have also noted in section 1, paragraph 3 that ML and AI approaches have been shown to accurately simulate net physical processes (for example rainfall-runoff modeling), so might be considered physics-based to some extent, but these approaches have seen only limited application in the context of future snow prediction so do not represent a viable alternative to existing approaches without further development.

The referee notes that the central philosophical argument of the paper is implied but not stated and suggests that a more transparent presentation of the paper's argument and its philosophical history might better enable readers to track the contribution of the paper in the broader development of the field of snow modeling. We agree that the manuscript would be enhanced by including or further illuminating these elements. To address this, we have added a paragraph to the introduction that directly states the central argument or the manuscript and provides some conceptual context.

We follow up on our central argument in the discussion by addressing the accuracy of our approach relative to existing large-extent snow products. We show that by modeling at higher spatial resolution and emphasizing fine-scale topographic effects (elevation, aspect, slope) but neglecting other processes (e.g. vegetation) we are able to better capture spatiotemporal snowpack variability across a large range of hydroclimates than existing, more comprehensive, coarser resolution products. This is even true compared to a study that used a combination of modeled, remotely sensed, and observed data to predict SWE (Guan et al., 2013). Furthermore, comparison with an average performance site (Fig 3) shows that our approach provides robust model performance on daily scales and in the context of interannual variability.

Another component of our central argument was that the SnowClim model is more computationally efficient than existing process-based approaches. This is difficult to assess directly given the lack of documentation of model run times and associated computational resources and also the difficulty of a meaningful model to model comparison. However, a handful of references in the literature to model run times suggest that SnowClim is more computationally efficient. Running SnowClim over the full western US domain for the historical period (13 years) took ~2.5 days on a standalone server (Dell Poweredge R730) with 34 cores and 128GB RAM. In contrast, Wrzesian et al., (2018) note that running WRF at 9 km resolution over North America on supercomputers for a single year took "~1.8 million core hours and months of real time". The run time per site year of the SnowClim model was less than 0.03 seconds when run across the 170 SNOTEL sites for 13 years using a single core (we have added this to section 3.4). This is greater than the run time per site year for a temperature index model (0.01 seconds) and less than an intermediate complexity energy balance snow model (0.03 to 0.06 seconds) found by Magnusson et al., (2015).

A more specific point begins at line 180 in the characterization of "single" layer snowpack models and shortcomings, particular with respect to the UEB model. The UEB model deals with the issue in two ways. During non-melt periods, it simply uses a conduction estimate based on a high frequency and a low frequency component – essentially using a Fourier decomposition approach to calculating the conduction rather than a finite difference approach – which would use "layers". UEB also makes further computations for heat transfer when the surface is melting. Insofar as you would like to emphasize the physical basis of snowclim, you may want to reconcile the "tax" approach and framing with the physical basis of conduction in the frequency domain. As it stands, when the surface is cooling, it looks like only a portion (as low as 10%) of the net flux at the surface is being removed from the pack – the remainder isn't accounted for at all. It is not clear what errors this accounts for, and you may want to add some content (text and appendix figures?) to the paper to explain how equations 16 work. There are at least 4 parameters involved, all of which are calibration parameters. This seems potentially important, but without some validation against temperature data, its difficult to tell.

We agree that the tax approach could benefit from additional explanation and evaluation since it is a new method. To this end, we have performed a comparison of our modeled cold content with and without the tax approach and observed cold content at two sites in the Niwot Ridge Long Term Ecological Research site (LTER), Colorado, an alpine site (Saddle) and a subalpine site (C1). The cold content observations are presented in Jennings et al., (2018).

For the comparison, we ran SnowClim at each of the sites using hourly observed climate forcings from the sites (Jennings et al., 2021) for water years 2001-2013. The model was run first as it is reported in the manuscript. Then, an additional run was performed in which the tax approach was turned off. The goal of this experiment was to evaluate how well the model (as presented in the manuscript) simulates cold content relative to observations and to determine whether the tax approach improved cold content simulations compared to the same model without the tax approach. The same parameters from the western US run were used for modeling these sites, the model was not recalibrated. Simulated cold content was smoothed using a two week moving mean to provide a more robust comparison to observations, which were collected at weekly to monthly intervals. Peak (minimum) cold content values were averaged across years to get a peak cold content value for each site and model run and these were compared with the peak cold content values from observations reported in Jennings et al., (2018), Table 1.

Despite scale discrepancies between point observations and 210 m grid cells, SnowClim peak cold content compares well with observations when the tax approach is used. In contrast, when the tax approach is not used and all negative energy fluxes are directly added to the pack cold content, cold content becomes an order of magnitude greater than the observations.

Table A1. Comparison of observed, simulated, and simulated without tax peak cold content at two sites.

| Site | Obs Peak CC | SnowClim Peak CC w/ tax approach | SnowClim Peak CC w/o tax approach |
|------|-------------|----------------------------------|------------------------------------|
| Subalpine (C1) | -2.5 MJ/m$^2$ | -4.1 MJ/m$^2$ | -27.0 MJ/m$^2$ |
| Alpine (Saddle) | -6.5 MJ/m$^2$ | -8.7 MJ/m$^2$ | -161.0 MJ/m$^2$ |

We have added this analysis as Appendix A in the revised manuscript for those seeking more information regarding the tax approach and have referenced this analysis in section 2.2.6 of the revised manuscript.

Jennings, K., T. Kittel, N. Molotch, and K. Yang. 2021. Infilled climate data for C1, Saddle, and D1, 1990 - 2019, hourly. ver 2. Environmental Data Initiative. https://doi.org/10.6073/pasta/0bf785f2f77c3f558f633853a4465404 (Accessed 2022-02-15).

Note that the snow surface temperature calculation in UEB takes into account this conduction heat flux from the surface along with the several other fluxes at the surface (e.g. solar, longwave, latent, sensible) to find the T at which all fluxes balance. This was not a particularly computationally burdensome iteration and the uncertainty cited by Raleigh (2013) was in parameter estimates for conduction, which Snoclim just does not consider. In the broad scheme of uncertainties for energy balance, I don't know that these are big numbers, but the criticisms in line 189-190, which are drawn mostly from Raleigh (2013) are not really warranted and don't bear repeating in the context of this paper. There remains some chance that the UEB calculations might be nearly as parsimonious and generate less uncertainty. No assessment has been done in this paper to evaluate these issues.

We thank the referee for this additional insight regarding UEB's method of calculating snow surface temperature. Based on these comments and those of the other referee, we have revised the text in the second paragraph of section 2.2.6, including removing the reference to Raleigh et al., (2013).

A thorough comparison of the present model with alternate models (specifically those using other approaches to snow surface temperature) could be a valuable effort. However, given the difficulty of creating a robust model comparison, we feel that such an effort is beyond the scope of the current work. In a future iteration of the SnowClim model we would be interested in further evaluating the pros and cons of different approaches to solving for snow surface temperature.

Reference (other than already cited in the paper):

Fleming, S. W., Watson, J. R., Ellenson, A., Cannon, A. J., & Vesselinov, V. C. (2021). Machine learning in Earth and environmental science requires education and research policy reforms. Nature Geoscience, 14(12), 878-880.